# Revisiting Weight Regularization for Low-Rank Continual Learning

**Yaoyue Zheng**[1,2,3]    **Yin Zhang**[4,2,3]    **Joost van de Weijer**[2,3]    **Gido M van de Ven**[5]
**Shaoyi Du**[1]    **Xuetao Zhang**[1]    **Zhiqiang Tian**[6*]

[1] State Key Laboratory of Human-Machine Hybrid Augmented Intelligence
  Institute of Artificial Intelligence and Robotics, Xi'an Jiaotong University, Shaanxi, China
[2] Computer Vision Center, Barcelona, Spain
[3] Universitat Autònoma de Barcelona, Barcelona, Spain
[4] School of Instrument Science and Engineering, Harbin Institute of Technology, Harbin, China
[5] Bernoulli Institute, University of Groningen, the Netherlands
[6] School of Software Engineering, Xi'an Jiaotong University, Shaanxi, China

## Abstract

Continual Learning (CL) with large-scale pre-trained models (PTMs) has recently gained wide attention, shifting the focus from training from scratch to continually adapting PTMs. This has given rise to a promising paradigm: parameter-efficient continual learning (PECL), where task interference is typically mitigated by assigning a task-specific module during training, such as low-rank adapters. However, weight regularization techniques, such as Elastic Weight Consolidation (EWC)-a key strategy in CL-remain underexplored in this new paradigm. In this paper, we revisit weight regularization in low-rank CL as a new perspective for mitigating task interference in PECL. Unlike existing low-rank CL methods, we mitigate task interference by regularizing a shared low-rank update through EWC, thereby keeping the storage requirement and inference costs constant regardless of the number of tasks. Our proposed method EWC-LoRA leverages a low-rank representation to estimate parameter importance over the full-dimensional space. This design offers a practical, computational- and memory-efficient solution for CL with PTMs, and provides insights that may inform the broader application of regularization techniques within PECL. Extensive experiments on various benchmarks demonstrate the effectiveness of EWC-LoRA, achieving a stability-plasticity trade-off superior to existing low-rank CL approaches. These results indicate that, even under low-rank parameterizations, weight regularization remains an effective mechanism for mitigating task interference. Code is available at: https://github.com/yaoyz96/low-rank-cl.

## 1 Introduction

Continual Learning (CL) has emerged as a rapidly growing research area, aiming to enable machine learning systems to acquire new knowledge without forgetting previously learned concepts (Parisi et al., 2019). This ability plays a crucial role in addressing real-world problems (Shaheen et al., 2022; Wang et al., 2024a) where data distributions are constantly changing. Ideally, a CL model should be able to maintain stable performance across all previously encountered tasks. A significant decline in performance on previous tasks after learning new ones is known as *catastrophic forgetting* (McCloskey & Cohen, 1989; Ratcliff, 1990), which typically arises from task interference.

With the rise of large-scale pre-trained models (PTMs) (Bommasani, 2021; Awais et al., 2025), the research focus in CL has shifted from training models from scratch to continually adapting these powerful models (Ostapenko et al., 2022; Yang et al., 2025). This trend is driven by the impressive transferability and robustness of PTMs, and a growing body of work has shown promising results in PTM-based continual adaptation. A particularly popular paradigm is parameter-efficient continual

---

*Corresponding author.

learning (PECL) (Qiao & Mahdavi, 2024), in which the PTM is typically kept frozen and augmented with lightweight modules such as prompts (Wang et al., 2022c; Smith et al., 2023), adapters (Ermis et al., 2022; Gao et al., 2024), or low-rank adaptations (LoRA) (Liang & Li, 2024; Wu et al., 2025). The predominant strategy in these works is to prevent task interference by assigning task-specific modules during training—either structurally isolated adapters or LoRA modules, or prompts that provide task-specific conditioning at the feature level.

On the other hand, weight regularization, as a key continual learning strategy, remains underexplored in the era of continual learning with PTMs. A canonical example is Elastic Weight Consolidation (EWC) (Kirkpatrick et al., 2017), which has played a central role in combating catastrophic forgetting in small-scale models (Schwarz et al., 2018; Liu et al., 2018; Ehret et al., 2020). Although effective for smaller models, EWC is difficult to apply to PTMs, as estimating parameter importance via the Fisher Information Matrix (FIM) is computationally expensive, requiring storage of a frozen copy of the old model and a Fisher matrix of equal size, resulting in a memory overhead three times that of the original model. Several studies have attempted to apply EWC to the fine-tuning of large language models (Xiang et al., 2023; Šliogeris et al., 2025). However, they typically fine-tune the model with a precomputed Fisher matrix that is fixed throughout training, making them impractical for CL.

In this paper, we adopt EWC as a canonical example to study weight regularization in low-rank CL, systematically analyzing key considerations for applying EWC within low-rank adaptations and proposing a feasible weight-regularization-based solution for low-rank CL. First, we revisit weight regularization in low-rank CL as a new perspective to mitigating catastrophic forgetting. Existing low-rank CL methods assign each task an independent LoRA module, constraining updates to subspaces that reduce interference with prior tasks. While effective, the addition of LoRA modules incurs storage overhead that scales linearly with the number of tasks. In contrast, we mitigate task interference by regularizing a shared low-rank update through EWC, rather than structurally isolating task-specific parameters, thereby keeping the storage requirement constant regardless of the number of tasks. Moreover, we provide the first systematic investigation of EWC in low-rank CL, and we theoretically and empirically demonstrate that a naïve integration of EWC with low-rank adaptation is suboptimal. To address this limitation, we propose a principled method to estimate parameter importance within the low-rank space. Our method leverages a full-dimensional FIM to reliably estimate the importance of parameters in low-rank updates. We empirically show that the proposed regularization on low-rank matrices achieves a better and more flexible stability-plasticity trade-off than existing low-rank methods (Wang et al., 2022c; Liang & Li, 2024; Wu et al., 2025), which are typically constrained to a fixed operating point.

Drawing on these insights, we propose **EWC-LoRA**, which updates the model via low-rank adaptation while leveraging the full-dimensional space FIM for weight regularization. EWC-LoRA does not explicitly fine-tune the full model or store model components for all previous tasks, thereby significantly reducing computational and memory overhead while enabling effective Fisher estimation, making it a resource-efficient solution for CL with PTMs. The main contributions of this work are as follows:

- We revisit weight regularization as a new perspective for mitigating catastrophic forgetting in low-rank CL. By exploiting the low-rank structure, we develop an efficient realization of EWC in PTMs. Specifically, by regularizing a shared LoRA module, EWC-LoRA maintains a constant memory footprint regardless of the number of tasks.

- We present the first systematic investigation of EWC in low-rank CL and propose estimating the FIM over the full-dimensional space to accurately capture parameter importance. Our analysis shows that naïvely integrating EWC with low-rank adaptation is suboptimal, while EWC-LoRA effectively regularizes learning and achieves a better stability-plasticity trade-off than existing low-rank CL methods.

- Extensive experiments across multiple benchmarks demonstrate that EWC-LoRA is effective, improving over vanilla LoRA by an average of 8.92%, while achieving comparable or even superior performance to state-of-the-art low-rank CL methods, with better computational and storage efficiency.

## 2 RELATED WORKS

**Continual Learning (CL).**   In contrast to standard supervised learning, which assumes that training data are independent and identically distributed (i.i.d.), CL focuses on training models on data streams that exhibit non-stationary and often continuous distribution shifts (Lesort et al., 2021). This departure from the i.i.d. assumption introduces the central challenge of catastrophic forgetting, where the model experiences significant performance degradation on previously learned tasks as new tasks are introduced (McCloskey & Cohen, 1989; Ratcliff, 1990). Following Van de Ven et al. (2022), CL can be categorized into three main scenarios: task-incremental (Gao et al., 2023), domain-incremental (Wang et al., 2024b), and class-incremental learning (Hersche et al., 2022). Among these, class-incremental learning can be considered the most challenging. *In this work, we adhere to the class-incremental learning setting, where the model must learn to distinguish between all classes encountered across all tasks* (Masana et al., 2022).

**Parameter-Efficient Continual Learning (PECL).**   PECL (Qiao & Mahdavi, 2024) has recently emerged as a promising paradigm in CL. It builds upon the idea of parameter-efficient fine-tuning (Houlsby et al., 2019; Hu et al., 2022; Jia et al., 2022), where a pre-trained model is kept frozen and a small number of learnable parameters are introduced to adapt to new tasks. To prevent task interference, existing PECL methods can be categorized into two types: (1) prompt-based methods, which typically provide each task with task-specific prompts to condition PTMs at feature level (Wang et al., 2022c;b; Smith et al., 2023), and (2) adapter- or low-rank adaptation-based methods, which typically insert task-specific lightweight modules during training, thereby providing isolation at the structural level (Gao et al., 2024; Liang & Li, 2024; Dou et al., 2024; Wu et al., 2025; Qian et al., 2025). Existing PECL works thus primarily focus on introducing task-specific modules to mitigate task interference, which often leads to increased memory and computational costs as the number of tasks grows. In contrast, weight regularization techniques have received little attention, and it remains unclear how they can be effectively applied in PECL—a setting that presents unique structural and optimization challenges compared to full-model tuning. For regularization-based low-rank CL methods, O-LoRA (Wang et al., 2023) mitigates task interference by enforcing geometric orthogonalization in the update subspace, thereby preserving the direction of parameter updates. *Within the context of low-rank CL, we revisit weight regularization as a means to mitigate task interference. In contrast to O-LoRA, we preserve the magnitude of updates through Fisher-based penalties, providing insights that may inform the broader application of regularization techniques within PECL—an area that remains underexplored in the current literature.*

**Elastic Weight Consolidation (EWC).**   EWC (Kirkpatrick et al., 2017) mitigates catastrophic forgetting in CL by penalizing changes to parameters that are deemed important for previous tasks, as quantified by the Fisher Information Matrix (FIM). As a canonical example of weight regularization techniques, EWC has inspired a series of follow-up studies that aimed to address its limitations and broaden its applicability. For example, Huszár (2018) analyzed its behavior beyond two tasks, while van de Ven (2025) investigated strategies for estimating the FIM in the context of CL. In the era of PTMs, Xiang et al. (2023) apply EWC during fine-tuning of a large language model (LLM), using a precomputed FIM to protect the knowledge acquired by the original model. Similarly, Šliogeris et al. (2025) employ EWC in the context of LLMs, estimating the FIM on a comprehensive benchmark to preserve domain knowledge. However, both studies rely on a precomputed FIM, which is kept fixed throughout training, making them unsuitable for our setting. Thede et al. (2024) briefly note the continued value of regularization in the context of PTMs, but they do not examine its detailed effect and do not combine regularization with low-rank adaptation. Wei et al. (2025) do combine EWC with low-rank adaptation, but they separately regularize each low-rank module, causing inaccurate Fisher estimation and suboptimal performance. *In this work, we conduct a focused investigation of EWC in PTMs-based CL. By leveraging a low-rank structure, we propose a practical approach for adapting EWC to PTM-based CL and demonstrate its effectiveness.*

## 3 METHODOLOGY

In this section, we review the necessary preliminaries, then discuss the structural and optimization challenges of applying EWC to low-rank adaptation, and finally present an overview of EWC-LoRA, highlighting its learning procedure and differences from existing low-rank CL methods.

### 3.1 PRELIMINARIES

**Notations.** In this paper, bold lowercase letters represent vectors, while bold uppercase letters denote matrices. The superscript $\top$ indicates the transpose of a matrix, and $\mathbb{E}[\cdot]$ stands for the expectation operator. Optimal values of variables are indicated with a superscript $*$.

**Problem Formulation.** We start with a pre-trained model parameterized by $\mathbf{W}_0$ and fine-tune it sequentially on a series of new tasks $\{\mathcal{T}_t\}_{t=1}^T$ with corresponding datasets $\{\mathcal{D}_t\}_{t=1}^T$. For each task $\mathcal{T}_t$, the model receives a batch of samples $\{x_k^t, y_k^t\}_{k=1}^{|\mathcal{D}_t|}$ drawn from $C$ classes, where $x_k^t$ and $y_k^t$ denote the input image and its corresponding label, respectively. After completing training on $\mathcal{T}_t$, the model is evaluated on all so-far encountered tasks $\mathcal{T}_{1:t}$. The objective is to learn parameters $\mathbf{W}$ that generalize well across all tasks so far, without storing any past data. With the model parameterized by $\mathbf{W}$, the training loss function at task $\mathcal{T}_t$ is usually defined as:

$$\mathcal{L}_t(\mathbf{W}) \;=\; -\frac{1}{|\mathcal{D}_t|} \sum_{k=1}^{|\mathcal{D}_t|} \sum_{c=1}^{C} \mathbb{1}_{[y_k^t = c]} \log p_{\mathbf{W}}(y = c \mid x_k^t) \tag{1}$$

**Elastic Weight Consolidation.** Following (Huszár, 2018), we maintain a single penalty term that approximates the combined effect of all previous tasks, preventing the double-counting inherent in the multi-penalty approach. When learning on task $\mathcal{T}_t$, we approximate the posterior $p(\mathbf{W}|\mathcal{D}_{1:t-1})$ using the Laplace approximation (MacKay, 1992), forming a Gaussian distribution $\mathcal{N}(\mathbf{W}; \mathbf{W}_{t-1}^*, (\mathbf{F}_{t-1}^{\mathrm{cum}})^{-1})$, where $\mathbf{W}_{t-1}^*$ denotes the optimal parameters on task $\mathcal{T}_{1:t-1}$, and the accumulated Fisher matrix $\mathbf{F}_{t-1}^{\mathrm{cum}}$ serves as the precision matrix, reflecting the importance of each parameter for retaining knowledge from all previous tasks. To improve computational efficiency, EWC assumes parameter independence and retains only the diagonal elements of the Fisher matrix. During training on $\mathcal{T}_t$, the loss function in Eq. 1 is augmented with a quadratic penalty term that constrains important parameters to remain close to their previously learned values:

$$\mathcal{L}_t'(\mathbf{W}) \;=\; \mathcal{L}_t(\mathbf{W}) \;+\; \frac{\lambda}{2}(\mathbf{W} - \mathbf{W}_{t-1}^*)^\top \operatorname{diag}(\mathbf{F}_{t-1}^{\mathrm{cum}})(\mathbf{W} - \mathbf{W}_{t-1}^*) \tag{2}$$

where $\lambda$ is a hyperparameter that controls the relative importance of the new task compared to the old one(s). $\operatorname{diag}(\mathbf{F}_{t-1}^{\mathrm{cum}})$ denotes the diagonal matrix formed from the diagonal elements of the accumulated Fisher matrix. Hereafter, unless otherwise stated, $\mathbf{F}$ refers to the diagonal Fisher matrix, with the $\operatorname{diag}(\cdot)$ omitted for simplicity.

**Low-rank Adaptation (LoRA).** When fine-tuning a model, LoRA (Hu et al., 2022) restricts changes to the parameters to lie in a low-rank subspace. Suppose $\mathbf{W}, \mathbf{W}_0 \in \mathbb{R}^{d_O \times d_I}$ are the adapted and pre-trained weight matrix, respectively. LoRA expresses the weight update $\Delta \mathbf{W} = \mathbf{W} - \mathbf{W}_0$ as the product of two learnable matrices $\mathbf{A} \in \mathbb{R}^{d_O \times r}$ and $\mathbf{B} \in \mathbb{R}^{r \times d_I}$, with $r \ll \min(d_I, d_O)$. Thus, the adapted weight matrix can be represented as $\mathbf{W} = \mathbf{W}_0 + \Delta \mathbf{W} = \mathbf{W}_0 + \mathbf{AB}$. During fine-tuning, the pre-trained weights $\mathbf{W}_0$ are frozen and only the parameters of $\mathbf{A}$ and $\mathbf{B}$ are trainable.

### 3.2 EWC WITH LOW-RANK ADAPTATION

The core idea of EWC in Eq. 2 lies in the second term, which measures how far the current parameters deviate from the previously learned ones. Directly applying EWC entails fine-tuning all parameters, as well as preserving both a frozen copy of the old model and a Fisher matrix of the same size. However, in the case of large PTMs, this is often not feasible. To reduce the number of trainable parameters and improve efficiency, we represent the weight update as $\Delta \mathbf{W} = \mathbf{W}_t - \mathbf{W}_{t-1}^* = \mathbf{AB}$ via low-rank decomposition.

To regularize low-rank matrices, a straightforward way is to compute individual Fisher matrices for $\mathbf{A}$ and $\mathbf{B}$, and apply regularization to each accordingly. This is the approach of Wei et al. (2025). However, focusing solely on the individual low-rank matrices ignores the interaction between $\mathbf{A}$ and $\mathbf{B}$, which is problematic because each element of the update $\Delta \mathbf{W}$ depends on their joint product, i.e., $\Delta \mathbf{W}_{ij} = \sum_{k=1}^r \mathbf{A}_{ik} \mathbf{B}_{kj}$. In Appendix A.1.1, we mathematically prove that regularization performed separately in the low-rank space generally diverges from that performed in the full-dimensional space. Another way to regularize low-rank matrices is to precompute the

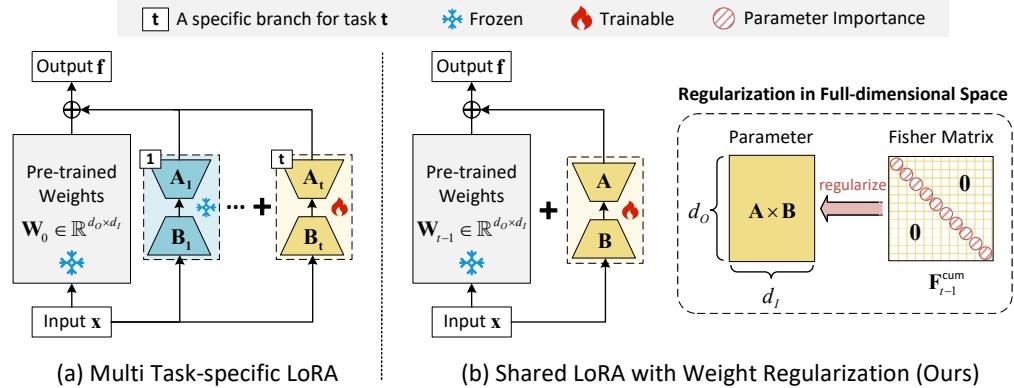

Figure 1: Overview of learning task $\mathcal{T}_t$ at a specific layer of the ViT model. **(a)** Prior low-rank CL methods structurally isolate task-specific LoRA parameters by adding a new LoRA branch for each task. **(b)** The proposed EWC-LoRA employs a shared LoRA module that is learned across all tasks and regularized according to parameter importance measured by a Fisher Information Matrix, which is updated after learning each task.

Fisher matrix on the pre-trained model and then apply the regularization to the update $\Delta\mathbf{W} = \mathbf{AB}$ using this fixed Fisher matrix (Xiang et al., 2023). However, this estimation also includes directions associated with the frozen $\mathbf{W}_0$, which can introduce noise in measuring sensitivity to the loss. This issue is further illustrated in Appendix A.1.2. In Table 1, we can observe that the two naïve methods for regularizing low-rank matrices result in suboptimal performance.

To address the above limitations, we propose updating parameters in the low-rank space while regularizing over the full-dimensional subspace of $\mathbf{W}$ spanned by $\mathbf{A}$ and $\mathbf{B}$, as illustrated in Figure 1 (b). This ensures that the penalty captures the true sensitivity of the model output to low-rank updates, rather than merely the local gradient magnitudes. Consequently, we reformulate Eq. 2 as:

$$\mathcal{L}'_t(\mathbf{A}, \mathbf{B}) = \mathcal{L}_t(\mathbf{A}, \mathbf{B}) + \frac{\lambda}{2} \operatorname{vec}(\mathbf{AB})^\top \mathbf{F}^{\text{cum}}_{t-1} \operatorname{vec}(\mathbf{AB}) \tag{3}$$

where $\operatorname{vec}(\mathbf{AB})$ is flattened $\mathbf{AB}$. This formulation enables regularization in the full-dimensional subspace $\Delta\mathbf{W}$, without requiring explicit storage of $\operatorname{vec}(\mathbf{AB})$. After training on $\mathcal{T}_t$, we estimate $\mathbf{F}_t$ and incorporate it into the previously accumulated Fisher matrix to obtain the updated Fisher $\mathbf{F}^{\text{cum}}_t$.

To estimate the Fisher matrix $\mathbf{F}_t$ at the optimal parameters $\mathbf{W}^*_t$ for task $\mathcal{T}_t$, we follow the definition in (Martens, 2020). Specifically, the $i$-th diagonal element of $\mathbf{F}_t$ is defined as:

$$F^{i,i}_t = \mathbb{E}_{x \sim \mathcal{D}_t}\left[\mathbb{E}_{y \sim p_{\mathbf{W}^*_t}}\left[\left(\frac{\partial \log p_{\mathbf{W}}(y|x)}{\partial w_i}\bigg|_{\mathbf{W}=\mathbf{W}^*_t}\right)^2\right]\right] \tag{4}$$

The outer expectation in Eq. 4 is computed over the current task data $\mathcal{D}_t$, while the inner expectation is approximated using the empirical Fisher for computational efficiency. In practice, this inner expectation can be estimated by taking the squared gradients of the log-likelihood with respect to $\mathbf{W}$, evaluated at $\mathbf{W}^*_t$. Since only the low-rank update $\Delta\mathbf{W}$ is trainable, the gradient with respect to $\mathbf{W}$ and $\Delta\mathbf{W}$ are identical. Consequently, the Fisher matrix is effectively computed in the $\Delta\mathbf{W}$-space. The equivalence between estimating the Fisher information in the $\mathbf{W}$-space and in the $\Delta\mathbf{W}$-space is established in Appendix A.1.3.

### 3.3 OVERVIEW OF EWC-LoRA

Figure 1 illustrates the difference between EWC-LoRA and existing state-of-the-art low-rank CL methods, in the context of learning task $\mathcal{T}_t$ at a specific layer of the Vision Transformer (ViT). For task $\mathcal{T}_t$, we initialize the shared LoRA branch, and the forward computation is given by: $\mathbf{f} = \mathbf{W}_{t-1}\mathbf{x} + \mathbf{AB}\mathbf{x}$. Specifically, $\mathbf{A}$ is zero-initialized, while $\mathbf{B}$ is drawn from a uniform distribution. During training on $\mathcal{T}_t$, only $\mathbf{A}$ and $\mathbf{B}$ are updated, while the base weights $\mathbf{W}_{t-1}$ keep frozen. After

Table 1: Comparison of different Fisher estimation strategies on CIFAR-100. "**+ Mem.**" indicates the additional memory required for Fisher estimation and regularization during training.

| Strategy | $\overline{A}_{10}$ | Avg. | Stability | Plasticity | + Mem. |
|---|---|---|---|---|---|
| w/o $\mathbf{F}$ | $82.99_{(0.84)}$ | $89.74_{(0.58)}$ | $87.56_{(0.09)}$ | $\mathbf{98.86}_{(0.09)}$ | 0 GB |
| Precomputed $\mathbf{F_W}$ | $83.87_{(0.21)}$ | $89.36_{(0.49)}$ | $93.15_{(0.45)}$ | $94.74_{(0.56)}$ | 1 GB |
| Separate $\mathbf{F_A}, \mathbf{F_B}$ | $86.41_{(0.69)}$ | $91.33_{(0.50)}$ | $94.23_{(0.46)}$ | $96.47_{(0.21)}$ | 4 GB |
| $\mathbf{F_{\Delta W}}$ (Ours) | $\mathbf{87.91}_{(0.57)}$ | $\mathbf{92.27}_{(0.39)}$ | $\mathbf{94.45}_{(0.59)}$ | $97.99_{(0.50)}$ | 6 GB |

completing task $\mathcal{T}_t$, the learned parameters are integrated to the base weight as $\mathbf{W}_t = \mathbf{W}_{t-1} + \mathbf{AB}$. For any sample from a previous task, the forward computation becomes $\mathbf{f} = \mathbf{W}_t\mathbf{x}$. During training, the update of the low-rank matrices $\mathbf{A}$ and $\mathbf{B}$ are regularized using the accumulated Fisher matrix $\mathbf{F}_{t-1}^{\text{cum}}$ from step $t-1$. Accordingly, EWC-LoRA maintains only two states after learning step $t$: (1) the updated model parameters $\mathbf{W}_t$ obtained from the current task $\mathcal{T}_t$, and (2) the accumulated Fisher matrix $\mathbf{F}_t^{\text{cum}}$, a diagonal matrix that aggregates information from all previous tasks $\mathcal{T}_{1:t}$. The dataset $\mathcal{D}_t$ and the task-specific Fisher matrix $\mathbf{F}_t$ can then be discarded. For clarity, we outline the learning procedure of EWC-LoRA in Algorithm 1, provided in Appendix A.2.1. For computational and memory efficiency, the only additional memory cost of EWC-LoRA comes from storing the FIM. However, due to the low-rank parameterization, EWC-LoRA incurs only a modest memory overhead while achieving higher computational efficiency than existing low-rank CL methods. A detailed efficiency analysis is provided in Section 4.2.

# 4 EXPERIMENTS

## 4.1 BENCHMARKS

**Datasets.** In line with existing continual learning methods (Wang et al., 2023; Liang & Li, 2024; Wu et al., 2025), we evaluate the performance of EWC-LoRA under two settings: (1) Four widely used CL vision benchmarks, including CIFAR-100 (Krizhevsky et al., 2009), DomainNet (Peng et al., 2019; Wang et al., 2022a), ImageNet-R (Hendrycks et al., 2021a; Wang et al., 2022b), and ImageNet-A (Hendrycks et al., 2021b); and (2) A standard language CL benchmark with five text classification datasets: AG News, Amazon Reviews, Yelp Reviews, DBpedia, and Yahoo Answers. Following Wang et al. (2023), we evaluate the models under three different task orders. The detailed task order is provided in Appendix A.3. CIFAR-100 consists of 100 natural image classes and is the most commonly used dataset in continual learning. DomainNet includes 345 classes across six diverse visual domains, making it a challenging multi-domain benchmark. Note that similar to Liang & Li (2024), DomainNet is split as a class-incremental learning problem. ImageNet-R contains 200 ImageNet classes (Deng et al., 2009) rendered with various artistic styles, introducing significant distribution shifts. ImageNet-A consists of 200 natural adversarial examples that are frequently misclassified by standard ImageNet-trained models. We follow the mostly used task splits for continual learning benchmarks: CIFAR-100 is divided into 10 tasks (10 classes per task); DomainNet into 5 tasks (69 classes per task); ImageNet-A into 10 tasks (20 classes each); ImageNet-R into 5, 10, and 20 tasks (with 40, 20, and 10 classes per task, respectively).

**Evaluation metrics.** We adopt accuracy as our evaluation metric, in line with standard practice (Lopez-Paz & Ranzato, 2017; Chaudhry et al., 2019). Let $A_{t,i}$ denote the classification accuracy on the $i$-th task after training on the $t$-th task. The average accuracy at learning step $t$ is defined as: $\overline{A}_t = \frac{1}{t}\sum_{i=1}^{t} A_{t,i}$. The overall average accuracy is then computed as the mean of all intermediate averages: Avg. $= \frac{1}{T}\sum_{t=1}^{T} \overline{A}_t$, where $T$ is the total number of tasks.

To further analyze a model's stability and plasticity and to enable intuitive comparisons across tasks, we propose measuring each metric on a normalized scale. Specifically, the stability score is defined as one minus the normalized forgetting:

$$\text{Stability} = 1 - \overline{F} = 1 - \frac{1}{T-1}\sum_{i=1}^{T-1} \frac{\max_{t<T} A_{t,i} - A_{T,i}}{\max_{t<T} A_{t,i}} \tag{5}$$

Table 2: Comparison results on CIFAR-100, DomainNet, ImageNet-R, and ImageNet-A (in %). **Bold** and underline indicate the highest and second-highest scores, respectively.

| Tasks | CIFAR-100 | | DomainNet | | ImageNet-R | | ImageNet-A | |
|---|---|---|---|---|---|---|---|---|
| Methods | $\overline{A}_{10}$ (↑) | Avg. (↑) | $\overline{A}_5$ (↑) | Avg. (↑) | $\overline{A}_{10}$ (↑) | Avg. (↑) | $\overline{A}_{10}$ (↑) | Avg. (↑) |
| Joint Train | $92.82_{(0.20)}$ | $95.41_{(0.05)}$ | $76.84_{(0.06)}$ | $81.25_{(0.05)}$ | $81.69_{(0.28)}$ | $86.25_{(0.09)}$ | $65.01_{(0.74)}$ | $74.10_{(0.44)}$ |
| Finetune | $79.09_{(1.53)}$ | $88.17_{(0.45)}$ | $65.57_{(0.20)}$ | $75.12_{(0.12)}$ | $60.42_{(1.64)}$ | $73.18_{(0.32)}$ | $32.85_{(1.53)}$ | $54.55_{(1.44)}$ |
| L2P | $83.18_{(1.20)}$ | $87.69_{(1.05)}$ | $70.26_{(0.25)}$ | $75.83_{(0.98)}$ | $71.26_{(0.44)}$ | $76.13_{(0.46)}$ | $42.94_{(1.27)}$ | $51.40_{(1.95)}$ |
| DualPrompt | $81.48_{(0.86)}$ | $86.41_{(0.66)}$ | $68.26_{(0.90)}$ | $73.84_{(0.45)}$ | $68.22_{(0.20)}$ | $73.81_{(0.39)}$ | $45.49_{(0.96)}$ | $54.68_{(1.24)}$ |
| CODA-Prompt | $86.31_{(0.12)}$ | $90.67_{(0.22)}$ | $70.58_{(0.53)}$ | $76.68_{(0.44)}$ | $74.05_{(0.41)}$ | $78.14_{(0.39)}$ | $45.36_{(0.78)}$ | $57.03_{(0.94)}$ |
| InfLoRA | $86.34_{(0.76)}$ | $91.33_{(0.48)}$ | $71.01_{(0.05)}$ | $77.75_{(0.03)}$ | $\underline{74.41}_{(0.63)}$ | $\underline{80.31}_{(0.60)}$ | $50.75_{(1.33)}$ | $64.36_{(1.01)}$ |
| SD-LoRA | $86.77_{(0.30)}$ | $90.96_{(0.30)}$ | $\underline{71.27}_{(0.14)}$ | $77.70_{(0.07)}$ | $72.93_{(2.76)}$ | $79.80_{(0.15)}$ | $55.23_{(0.94)}$ | $66.10_{(0.54)}$ |
| CL-LoRA | $\underline{87.65}_{(0.53)}$ | $\underline{92.25}_{(0.41)}$ | $71.06_{(0.31)}$ | $\underline{77.76}_{(0.29)}$ | $\mathbf{78.72}_{(0.44)}$ | $\mathbf{85.20}_{(0.45)}$ | $\underline{57.62}_{(0.89)}$ | $\mathbf{70.76}_{(0.63)}$ |
| BiLoRA | $85.99_{(0.49)}$ | $90.62_{(0.42)}$ | $69.75_{(0.23)}$ | $73.86_{(0.21)}$ | $74.28_{(0.92)}$ | $77.38_{(0.88)}$ | $51.05_{(0.74)}$ | $62.82_{(0.65)}$ |
| Vanilla LoRA | $82.99_{(0.84)}$ | $89.74_{(0.58)}$ | $69.79_{(0.11)}$ | $77.44_{(0.08)}$ | $64.87_{(0.73)}$ | $75.57_{(0.23)}$ | $40.01_{(1.32)}$ | $58.28_{(0.85)}$ |
| EWC-LoRA | $\mathbf{87.91}_{(0.57)}$ | $\mathbf{92.27}_{(0.39)}$ | $\mathbf{73.46}_{(0.16)}$ | $\mathbf{79.58}_{(0.10)}$ | $72.86_{(0.79)}$ | $78.95_{(0.86)}$ | $\mathbf{59.89}_{(0.26)}$ | $\underline{68.33}_{(0.67)}$ |

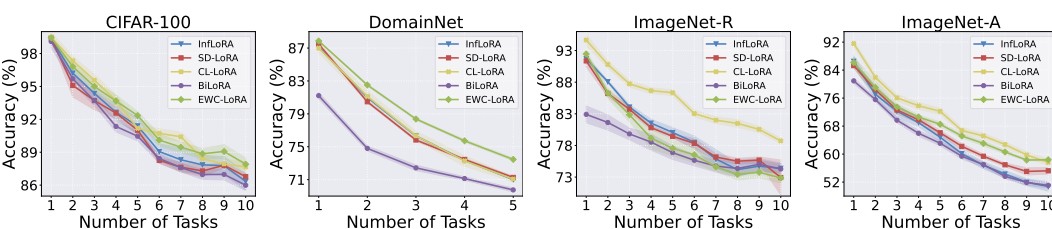

Figure 2: Task-wise performance comparison of different methods across various datasets.

where normalized forgetting $\overline{F}$ represents the relative drop in a task's performance from its peak to the end of continual learning. For each task $i$, plasticity is defined as the ratio between the model's performance on the task after learning it and the corresponding reference performance. The overall plasticity is defined as:

$$\text{Plasticity} = \frac{1}{T} \sum_{i=1}^{T} \frac{A_{i,i}}{A_{i,i}^{\text{ref}}} \qquad (6)$$

where $A_{i,i}^{\text{ref}}$ denotes the accuracy obtained by fine-tuning a model exclusively on task $i$. This normalization facilitates intuitive comparison across tasks and methods.

**Implementation details.** Following prior works (Wang et al., 2022c; 2023; Smith et al., 2023; Liang & Li, 2024), we adopt the ViT-B/16 backbone (Dosovitskiy et al., 2020) pretrained on ImageNet-21K in a supervised manner as the initialization for vision models, and use the pre-trained encoder-decoder T5-large and LLaMA-3.2-1B-Instruct for language tasks. To facilitate comparison, all methods are implemented within a unified framework. We align the experimental setup with that of InfLoRA (Liang & Li, 2024), fine-tuning the model with the Adam optimizer using hyperparameters $\beta_1 = 0.9$, $\beta_2 = 0.999$. For each comparison method, we report the best results using the hyperparameters provided by the authors whenever available. In cases where such configurations are not released, we apply a unified set of hyperparameters that has been validated to perform reliably across all methods, ensuring consistency in our experimental setup. To better contextualize performance, we report both an upper target (*Joint Train*) and a lower target (*Finetune*). The upper target jointly trains on all tasks, while the lower target sequentially trains on tasks without any forgetting mitigation. For all experiments, we perform five runs with different seeds and report the average and standard deviation of the results.

## 4.2 MAIN RESULTS

**Comparison with Various CL Baselines.** We benchmark EWC-LoRA against state-of-the-art PTM-based continual learning methods, including the prompt-based methods L2P (Wang et al., 2022c), DualPrompt (Wang et al., 2022b), and CODA-Prompt (Smith et al., 2023), as well as the LoRA-based methods InfLoRA (Liang & Li, 2024), SD-LoRA (Wu et al., 2025), CL-LoRA (He

et al., 2025), and BiLoRA (Zhu et al., 2025). We report results on 10 sequential tasks for CIFAR-100, ImageNet-R, and ImageNet-A, and on 5 tasks for DomainNet. The results are summarized in Table 2. From the table, we observe that EWC-LoRA achieves the highest final accuracy on three out of four datasets. On average, across all four datasets, EWC-LoRA outperforms vanilla LoRA by a substantial margin of +8.92%. EWC-LoRA even surpasses other LoRA-based methods that use task-specific low-rank modules, highlighting its effectiveness. On DomainNet, EWC-LoRA reduces the gap with the upper target Joint Training significantly from 5.57% of SD-LoRA to 3.38% for EWC-LoRA. On the challenging ImageNet-A, it reduces the gap considerably from 7.39% for CL-LoRA to only 5.12% for EWC-LoRA. Figure 2 illustrates the task-wise performance of LoRA-based methods. We observe that EWC-LoRA consistently outperforms other methods throughout the entire task sequence on most datasets, while also exhibiting lower standard deviations, indicating greater stability. We further evaluate EWC-LoRA on LLMs for language CL tasks and compare it with the LoRA-based methods O-LoRA (Wang et al., 2023) and TreeLoRA (Qian et al., 2025). The average accuracy is reported in Table 3. From the table, EWC-LoRA achieves comparable or even superior performance across the three task orders, demonstrating its effectiveness and applicability to LLMs.

Table 3: Comparison results (reported as average accuracy (%)) on the standard language CL benchmark with the two pretrained models. Details about the task order are provided in Appendix A.3.1.

| Backbone | Method | Order-1 | Order-2 | Order-3 | Avg. |
|---|---|---|---|---|---|
| T5-large | O-LoRA | 75.69 | 74.92 | **74.40** | 75.01 |
| | EWC-LoRA | **78.01** | **76.85** | 74.30 | **76.39** |
| LLaMA-3.2-1B-Instruct | O-LoRA | 56.96 | 55.74 | 67.32 | 60.01 |
| | TreeLoRA | 58.54 | 56.96 | 65.42 | 60.30 |
| | EWC-LoRA | **61.17** | **60.47** | **67.61** | **63.08** |

**Analysis on Stability and Plasticity.** We evaluated both stability and plasticity for different low-rank CL methods across the four datasets. Stability reflects the ability of the model to retain previously learned knowledge, while plasticity measures its ability to adapt to new tasks. The results are reported in Table 4. As expected, Vanilla LoRA typically exhibits the highest plasticity but the lowest stability, since it employs no strategy to mitigate catastrophic forgetting. We observe that InfLoRA emphasizes stability, while SD-LoRA favors plasticity. On CIFAR-100, EWC-LoRA matches InfLoRA in stability while achieving higher plasticity; on ImageNet-A, it matches SD-LoRA in plasticity while providing greater stability. On DomainNet and ImageNet-R, while other methods struggle to balance stability and plasticity, EWC-LoRA can achieve this trade-off effectively by tuning the regularization strength $\lambda$.

Table 4: Stability ($\uparrow$) and plasticity ($\uparrow$) scores of different low-rank CL methods, reflecting how well each model retains previous knowledge and adapts to new tasks. We report the normalized form of the two metrics, which is independent of the absolute performance on the dataset.

| Tasks | CIFAR-100 | | DomainNet | | ImageNet-R | | ImageNet-A | |
|---|---|---|---|---|---|---|---|---|
| Methods | Stability | Plasticity | Stability | Plasticity | Stability | Plasticity | Stability | Plasticity |
| Vanilla LoRA | $87.56_{(0.09)}$ | $98.86_{(0.09)}$ | $81.29_{(0.34)}$ | $97.94_{(0.16)}$ | $78.63_{(1.38)}$ | $99.57_{(0.35)}$ | $85.56_{(1.82)}$ | $97.56_{(0.77)}$ |
| InfLoRA | $94.84_{(0.52)}$ | $95.80_{(1.00)}$ | $83.80_{(0.37)}$ | $97.34_{(0.29)}$ | $92.69_{(0.77)}$ | $97.33_{(0.46)}$ | $88.63_{(3.39)}$ | $72.69_{(3.13)}$ |
| SD-LoRA | $91.85_{(0.60)}$ | $98.24_{(0.25)}$ | $82.45_{(0.19)}$ | $98.69_{(0.15)}$ | $91.78_{(0.90)}$ | $95.61_{(0.32)}$ | $88.61_{(1.05)}$ | $92.13_{(2.77)}$ |
| EWC-LoRA | $94.45_{(0.59)}$ | $97.99_{(0.50)}$ | $91.51_{(0.36)}$ | $93.83_{(0.23)}$ | $95.62_{(0.42)}$ | $93.23_{(0.34)}$ | $89.52_{(1.14)}$ | $92.78_{(1.91)}$ |

Specifically, for the results in Table 2 and 4, a unified regularization strength is used across datasets. With $\lambda = 10^7$, EWC-LoRA consistently achieves a favorable balance between stability and plasticity, without requiring dataset-specific tuning. To explore the trade-off between the two metrics, we show stability-plasticity curves for different values of the regularization strength, as illustrated in Figure 3a. Unlike other methods that typically exhibit fixed performance, EWC-LoRA provides a clear and controllable trade-off: smaller $\lambda$ promotes plasticity at the cost of stability, while larger values enhance stability but reduce plasticity. This tunability enables EWC-LoRA to achieve competitive or superior performance across a wide range of trade-off points, demonstrating its robustness and adaptability. We also note that even when different methods achieve similar average accuracy,

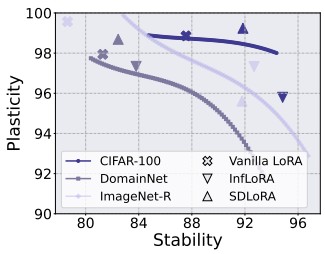 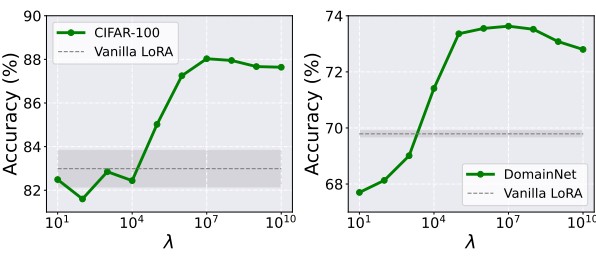

(a) Stability-Plasticity curves.  (b) Performance across a range of regularization strengths $\lambda$.

Figure 3: (a) Stability-Plasticity curves illustrating the trade-off between retaining previous knowledge and learning new tasks. (b) Performance across a range of regularization strengths $\lambda$ on CIFAR-100 and DomainNet, showing the effect of $\lambda$ on accuracy.

they can differ substantially in terms of stability and plasticity. This suggests that CL model evaluation should place greater emphasis on explicitly reporting stability and plasticity metrics.

**Memory footprint and training time.**   The additional memory and computational cost of EWC-LoRA compared to Vanilla LoRA comes only from the FIM. After each learning step, the LoRA parameters are merged into the backbone, so the total parameter size matches that of the pre-trained model. As a result, only one shared LoRA is required for new tasks, and the memory footprint remains constant regardless of the number of tasks. In contrast, other low-rank CL methods require maintaining separate LoRA parameters for each task or performing more complex computations, leading to a linear increase in memory usage and training time, or to higher computational cost as the number of tasks grows. EWC-LoRA incurs modest memory overhead during training while improving computational efficiency, achieving comparable or even superior performance.

Table 5: Comparison of different methods in terms of memory cost and training time. Memory usage is measured on the Quadro RTX 6000 GPU with a batch size of 128. Training time is reported as the average time required to train a single task.

| Methods | Memory | **CIFAR-100** | **DomainNet** | **ImageNet-R** | **ImageNet-A** |
|---|---|---|---|---|---|
| Vanilla LoRA | $\sim$ 18 GB | 10m22s $\pm$ 7s | 32m8s $\pm$ 44s | 13m32s $\pm$ 105s | 0m55s $\pm$ 21s |
| InfLoRA | $\sim$ 20 GB | 11m9s $\pm$ 4s | 42m25s $\pm$ 68s | 14m33s $\pm$ 107s | 1m16s $\pm$ 22s |
| SD-LoRA | $\sim$ 37 GB | 12m16s $\pm$ 86s | 34m18s $\pm$ 160s | 22m53s $\pm$ 232s | 1m56s $\pm$ 36s |
| EWC-LoRA | $\sim$ 24 GB | 10m31s $\pm$ 3s | 33m10s $\pm$ 50s | 13m42s $\pm$ 104s | 0m55s $\pm$ 21s |

The memory cost and training time are reported in Table 5. Memory usage is measured on two Quadro RTX 6000 GPUs with a batch size of 128. Training time is recorded as the average duration to train a single task, presented along with the standard deviation. Notably, the training time of EWC-LoRA is nearly identical to that of Vanilla LoRA, demonstrating that the additional computations introduced by Fisher estimation in the low-rank space incur relatively small overhead.

## 4.3  ABLATION STUDY

**Results Across Varied Task Lengths.**   We further examine the performance of different low-rank CL methods under varying task lengths. As a complementary study to Table 2, we split CIFAR-100 and ImageNet-R into 5-task and 20-task sequences, respectively. As reported in Table 6, the improvement is most evident on CIFAR-100, where EWC-LoRA achieves the highest overall accuracy among all methods. In contrast, the gains on ImageNet-R are more moderate, which may be attributed to the domain shift in ImageNet-R, potentially limiting the effectiveness of regularization-based approaches for continual adaptation. Additional results are provided in the Appendix A.3.

**Ablation on Regularization Strength $\lambda$.**   We evaluate performance under varying values of the regularization strength $\lambda$, ranging from 10 to $10^{10}$. The results are presented in Figures 3b. We use empirical Fisher in all experiments. From the figures, we observe that setting $\lambda$ to $10^7$ allows

Table 6: Final accuracy on CIFAR-100 and ImageNet-R for task lengths of 5 and 20 tasks.

| Tasks | CIFAR-100 | | ImageNet-R | |
|---|---|---|---|---|
| Methods | $\overline{A}_5$ | $\overline{A}_{20}$ | $\overline{A}_5$ | $\overline{A}_{20}$ |
| Vanilla LoRA | 87.15 | 74.78 | 70.15 | 56.17 |
| InfLoRA | 89.45 | 81.77 | **77.37** | 69.63 |
| SD-LoRA | 89.15 | 83.57 | 74.90 | **72.26** |
| EWC-LoRA | **89.98** | **85.46** | 76.36 | 70.18 |

Table 7: Best performance with Exact Fisher (500 random samples) and Empirical Fisher on CIFAR-100.

| Estimation | Exact (n=500) | Empirical |
|---|---|---|
| $\overline{A}_{10}$ | 88.28 | 87.91 |
| Avg. | 92.76 | 92.27 |
| Best $\lambda$ | $\lambda = 10^5$ | $\lambda = 10^7$ |
| Training Time | $\sim 14$m | $\sim 11$m |
| Memory Cost | $\sim 20$ GB | $\sim 24$GB |

EWC-LoRA to consistently achieve favorable performance in terms of accuracy, stability, and plasticity. This suggests that an appropriate balance between stability and plasticity can be effectively maintained without fine-grained tuning. This finding facilitates the use of a unified regularization strength across datasets, eliminating the need for dataset-specific tuning.

**Estimation of Fisher Information Matrix.** In the context of low-rank adaptation, we examine the trade-off between performance and the computational cost of estimating the Fisher Information Matrix, as discussed by van de Ven (2025). In general, the Exact Fisher outperforms the Empirical Fisher, requiring a smaller regularization strength, but higher computational costs. To reduce computational costs, the Exact Fisher can be computed using only a small batch of data, which still yields superior results compared to the Empirical Fisher while significantly reducing computational overhead. We use 500 randomly selected samples to estimate the Exact Fisher, and the comparison results are shown in Table 7. We report both the training time for a single task and the corresponding memory usage. Additional results comparing different strategies for estimating the Fisher matrix are provided in Appendix A.3.

**Different Fisher Estimation Strategies.** We compare three different Fisher estimation strategies, based on the discussion in Section 3.2. "Precomputed $\mathbf{F_W}$" refers to using a precomputed FIM in full parameter space $\mathbf{W}$ to preserve prior knowledge. Instead of estimating over a large-scale dataset, we consider a dataset-based Fisher, computed using the entire dataset that will be learned sequentially. "Separate $\mathbf{F_A}, \mathbf{F_B}$" denotes estimating the Fisher separately for the low-rank matrices and regularizing them accordingly, while "$\mathbf{F_{\triangle W}}$" denotes estimating the Fisher in the full-dimensional space as $\mathbf{W}$. The results are shown in Table 1. We observe that the precomputed Fisher results in the lowest plasticity, which may be caused by undesirable sensitivity arising from the frozen weights. Moreover, applying separate regularization in the low-rank space also improves performance but has noticeable drawbacks in terms of plasticity. This may be due to the joint contribution of the low-rank factors, which imposes stronger constraints on the parameters.

## 5 CONCLUSION AND DISCUSSION

In this work, we revisit weight regularization in low-rank continual learning (CL) as a means to mitigate catastrophic forgetting. Using Elastic Weight Consolidation (EWC) as a canonical example, we discuss the main considerations about applying regularization in the low-rank space and propose EWC-LoRA, a computational- and memory-efficient solution for low-rank CL with large pre-trained models. This work aims to offer insights that may guide the broader application of regularization techniques in parameter-efficient continual learning.

A key limitation of regularization-based low-rank CL methods lies in two aspects. The first is the degradation in the accuracy of Fisher estimation as the task sequence grows longer. This can be alleviated by rehearsal-based Fisher estimation, as discussed in Wu et al. (2024) and our Appendix. The second is their sensitivity to dataset complexity. When combined with regularization techniques, it is important to carefully allocate the low-rank learnable space for each task. This sensitivity also helps explain why, on ImageNet-R, EWC-LoRA shows lower plasticity than methods based on task-specific modules. Consequently, a promising direction for future work is to investigate the performance of regularization techniques in domain-incremental settings and on more complex continual learning tasks.

## 6 ACKNOWLEDGMENTS

This paper was supported by European Union's Horizon Europe research and innovation programme under grant agreement number 101214398 (ELLIOT). We acknowledge project Grant AIA2025-163919-C52 and PID2022-143257NB-I00 funded by MICIU/AEI/ 10.13039/501100011033 and FEDER and by the European Union - Next Generation EU, grant number SDC00725000001. Yaoyue Zheng acknowledges the China Scholarship Council (CSC) No.202406280387. This work was supported by the National Natural Science Foundation of China under Grant Nos. U24A20252 and 62125305, Fundamental and Interdisciplinary Disciplines Breakthrough Plan of the Ministry of Education of China under Grant No. JYB2025XDXM504, and Guangdong Major Project of Basic and Applied Basic Research under Grant No. 2023B0303000009.

## 7 REPRODUCIBILITY STATEMENT

The theoretical analysis and complete proofs of the main results are provided in Appendix A.1. The implementation details of our method, including model architecture, training procedures, and hyperparameters, are described in Section 3 of the main paper and Appendix A.2. All datasets used in our experiments are publicly available. Code is available at: `https://github.com/yaoyz96/low-rank-cl`.

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

# A    APPENDIX

## A.1    THEORETICAL ANALYSIS

**Setup.**   In low-rank adaptation, the adapted weight matrix $\mathbf{W} \in \mathbb{R}^{d_O \times d_I}$ is reparameterized as $\mathbf{W} = \mathbf{W}_0 + \Delta\mathbf{W} = \mathbf{W}_0 + \mathbf{AB}$, where $\mathbf{W}_0$ denotes the pre-trained weight, which remains frozen during fine-tuning. The trainable parameters are the low-rank matrices $\mathbf{A} \in \mathbb{R}^{d_O \times r}$ and $\mathbf{B} \in \mathbb{R}^{r \times d_I}$, such that $\Delta\mathbf{W} = \mathbf{AB}$ is the only trainable update to the model. Note that the learned LoRA modules on the previous task are merged back into the base weights before starting the new task. The optimal weight matrix is indicated as $\mathbf{W}^*$.

### A.1.1    REGULARIZATION UNDER DIFFERENT PARAMETERIZATIONS

**Proposition 1** *Let $\Delta\mathbf{W} \in \mathbb{R}^{d_O \times d_I}$ be a model parameter matrix factorized as $\Delta\mathbf{W} = \mathbf{AB}$, with $\mathbf{A} \in \mathbb{R}^{d_O \times r}$, $\mathbf{B} \in \mathbb{R}^{r \times d_O}$. Define the EWC regularization term in the full-space as $\mathcal{R}_\mathbf{W}$, and in the low-rank parameter space as $\mathcal{R}_{\mathbf{A},\mathbf{B}}$. Under general conditions, $\mathcal{R}_{\Delta\mathbf{W}} \not\equiv \mathcal{R}_{\mathbf{A},\mathbf{B}}$.*

***Proof.***   Let $\text{vec}(\cdot)$ denote the vectorization operator, and $\otimes$ the Kronecker product. The following identity holds:
$$\text{vec}(\mathbf{AB}) = (\mathbf{B}^\top \otimes \mathbf{I}_{d_O})\,\text{vec}(\mathbf{A}) = (\mathbf{I}_{d_I} \otimes \mathbf{A})\,\text{vec}(\mathbf{B})$$

For simplicity, we define $\mathbf{a} = \text{vec}(\mathbf{A}) \in \mathbb{R}^{d_O r}$, $\mathbf{b} = \text{vec}(\mathbf{B}) \in \mathbb{R}^{r d_I}$. Define the Jacobians:
$$J_A(\mathbf{B}) := \mathbf{B}^\top \otimes \mathbf{I}_{d_O} \in \mathbb{R}^{d_O d_I \times d_O r}, \quad J_B(\mathbf{A}) := \mathbf{I}_{d_I} \otimes \mathbf{A} \in \mathbb{R}^{d_O d_I \times r d_I}$$

By the standard vectorization identity:
$$\text{vec}(\Delta\mathbf{W}) = J_A(\mathbf{B})\mathbf{a} = J_B(\mathbf{A})\mathbf{b}$$

The Fisher Information Matrix estimated in the full-dimensional space is defined as:
$$\mathbf{F}_\mathbf{W} = \mathbb{E}_{x \sim \mathcal{D}}\left[\text{vec}(\nabla_\mathbf{W}\mathcal{L})\,\text{vec}(\nabla_\mathbf{W}\mathcal{L})^\top\right]$$

Although the full-space regularization $\mathcal{R}_\mathbf{W}$ can be expressed equivalently using either the $\mathbf{A}$ or $\mathbf{B}$ Jacobian, the update directions for $\mathbf{A}$ and $\mathbf{B}$ remain coupled. Thus, we consider a first-order approximation of $\text{vec}(\Delta\mathbf{W})$ as a function of both $\mathbf{a}$ and $\mathbf{b}$:
$$\text{vec}(\Delta\mathbf{W}) \approx J_A(\mathbf{B})\Delta\mathbf{a} + J_B(\mathbf{A})\Delta\mathbf{b}$$

The EWC regularization term in the full-space is defined as:
$$\begin{aligned}
\mathcal{R}_\mathbf{W} &= \frac{1}{2}(\text{vec}(\mathbf{W}) - \text{vec}(\mathbf{W}^*))^\top \mathbf{F}_\mathbf{W}(\text{vec}(\mathbf{W}) - \text{vec}(\mathbf{W}^*)) \\
&= \frac{1}{2}\text{vec}(\Delta\mathbf{W})^\top \mathbf{F}_\mathbf{W}\,\text{vec}(\Delta\mathbf{W}) \\
&= \frac{1}{2}\Delta\mathbf{a}^\top J_A(\mathbf{B})^\top \mathbf{F}_\mathbf{W} J_A(\mathbf{B})\Delta\mathbf{a} + \frac{1}{2}\Delta\mathbf{b}^\top J_B(\mathbf{A})^\top \mathbf{F}_\mathbf{W} J_B(\mathbf{A})\Delta\mathbf{b} \\
&\quad + \Delta\mathbf{a}^\top J_A(\mathbf{B})^\top \mathbf{F}_\mathbf{W} J_B(\mathbf{A})\Delta\mathbf{b}
\end{aligned}$$

If Fisher regularization is applied separately to $\mathbf{A}$ and $\mathbf{B}$, we have two Fisher matrices:
$$\begin{aligned}
\mathbf{F}_\mathbf{A} &= \mathbb{E}_{x \sim \mathcal{D}}\left[\text{vec}(\nabla_\mathbf{A}\mathcal{L})\,\text{vec}(\nabla_\mathbf{A}\mathcal{L})^\top\right] \\
\mathbf{F}_\mathbf{B} &= \mathbb{E}_{x \sim \mathcal{D}}\left[\text{vec}(\nabla_\mathbf{B}\mathcal{L})\,\text{vec}(\nabla_\mathbf{B}\mathcal{L})^\top\right]
\end{aligned}$$

By the chain rule, the gradients in low-rank factor space are related to the full-space gradient by:
$$\begin{aligned}
\nabla_\mathbf{A}\mathcal{L} &= \frac{\partial\mathcal{L}}{\partial\mathbf{W}} \cdot \frac{\partial\mathbf{W}}{\partial\mathbf{A}} = \nabla_\mathbf{W}\mathcal{L}J_A(\mathbf{B})^\top \\
\nabla_\mathbf{B}\mathcal{L} &= \frac{\partial\mathcal{L}}{\partial\mathbf{W}} \cdot \frac{\partial\mathbf{W}}{\partial\mathbf{B}} = J_B(\mathbf{A})^\top \nabla_\mathbf{W}\mathcal{L}
\end{aligned}$$

Hence, the Fisher Information Matrices can be expressed as:

$$\mathbf{F_A} \approx J_A(\mathbf{B})^\top \mathbf{F_W} J_A(\mathbf{B}), \quad \mathbf{F_B} \approx J_B(\mathbf{A})^\top \mathbf{F_W} J_B(\mathbf{A})$$

The EWC regularization term in the low-rank parameter space is defined as:

$$\mathcal{R}_{\mathbf{A},\mathbf{B}} = \frac{1}{2}\Delta\mathbf{a}^\top \mathbf{F_A}\Delta\mathbf{a} + \frac{1}{2}\Delta\mathbf{b}^\top \mathbf{F_B}\Delta\mathbf{b}$$

The regularization term $\mathcal{R}_{\mathbf{A},\mathbf{B}}$ captures independent penalties in the factor space. Unless the cross-covariance term $\mathbf{F_{A,B}}$ is explicitly computed and retained, the interaction between $\mathbf{A}$ and $\mathbf{B}$ is ignored. Consequently, the two regularizations cannot be equal, except in special cases such as when only $\mathbf{A}$ or $\mathbf{B}$ is updated. In contrast, estimating $\mathbf{F_W}$ in the full-dimensional space avoids this issue and captures the true sensitivity of the model to perturbations. Therefore, regularization in the full space better preserves the true geometry and parameter importance induced by the low-rank update, providing a more faithful and effective constraint on $\mathbf{A}$ and $\mathbf{B}$.

### A.1.2 FISHER INFORMATION CONSISTENCY WITH THE TRAINABLE PARAMETER SPACE

**Proposition 2** *Consider a parameterization:*

$$\mathbf{W} = \mathbf{W}_0 + \mathbf{AB}$$

*where $\mathbf{W}_0$ is fixed and only $\mathbf{A}, \mathbf{B}$ are trainable. Let*

$$\theta = \begin{bmatrix} \mathrm{vec}(\mathbf{A}) \\ \mathrm{vec}(\mathbf{B}) \end{bmatrix}, \quad \mathbf{w} = \mathrm{vec}(\mathbf{W}).$$

*Then, the Fisher Information Matrix with respect to $\theta$ satisfies*

$$\mathbf{F}_\theta = \mathbf{J}^\top \mathbf{F_W} \mathbf{J},$$

*where $\mathbf{F_W}$ is the Fisher Information Matrix with respect to $\mathbf{w}$, and $\mathbf{J} = \partial\mathbf{w}/\partial\theta$ is the Jacobian of the reparameterization.*

*Proof.* Let $\mathcal{L}(\mathbf{W})$ denote the loss (negative log-likelihood). By the chain rule,

$$\nabla_\theta \mathcal{L} = \mathbf{J}^\top \nabla_\mathbf{W} \mathcal{L}$$

The Fisher Information Matrix with respect to $\theta$ is defined as:

$$\mathbf{F}_\theta = \mathbb{E}\left[ \nabla_\theta \log p \, \nabla_\theta \log p^\top \right]$$

Substituting the chain rule,

$$\mathbf{F}_\theta = \mathbb{E}\left[ (\mathbf{J}^\top \nabla_\mathbf{w} \log p)(\mathbf{J}^\top \nabla_\mathbf{w} \log p)^\top \right]$$
$$= \mathbf{J}^\top \mathbb{E}\left[ \nabla_\mathbf{w} \log p \, \nabla_\mathbf{w} \log p^\top \right] \mathbf{J}.$$

The term inside the expectation is precisely the Fisher Information Matrix with respect to $\mathbf{w}$, denoted $\mathbf{F_W}$. Hence we have:

$$\mathbf{F}_\theta = \mathbf{J}^\top \mathbf{F_W} \mathbf{J} \tag{7}$$

The Jacobian structure has been illustrated in Proposition 1. The Fisher Information Matrix is fundamentally an estimation of the curvature of the loss function with respect to the trainable parameters. Eq. 7 shows that the correct Fisher matrix for $\theta$ is obtained by projecting $\mathbf{F_W}$ into the subspace spanned by $\mathbf{A}$ and $\mathbf{B}$ via $\mathbf{J}$. Computing $\mathbf{F_W}$ directly in the full parameter space $\mathbf{W}$ introduces directions corresponding to the frozen $\mathbf{W}_0$, which are irrelevant for training. Thus, Fisher should be consistently defined in the trainable subspace.

### A.1.3 Constraining Low-Rank Updates via Full-Space Fisher Regularization

**Proposition 3** *Let the adapted weight matrix be:* $\mathbf{W} = \mathbf{W}_0 + \Delta\mathbf{W} = \mathbf{W}_0 + \mathbf{AB}$*, where* $\mathbf{W}_0$ *denotes the frozen pre-trained weights and* $\Delta\mathbf{W} = \mathbf{AB}$ *is the low-rank update. Then, the empirical Fisher Information Matrix* $\mathbf{F}_{\Delta\mathbf{W}}$ *is estimated by the squared gradients of the log-likelihood with respect to* $\mathbf{W}$*. Consequently, the Fisher regularization on* $\Delta\mathbf{W}$ *induces constraints on the update directions of the low-rank factors* $\mathbf{A}$ *and* $\mathbf{B}$*.*

***Proof.*** Since $\mathbf{W}_0$ is constant and only $\Delta\mathbf{W}$ is trainable, the loss function $\mathcal{L}$ depends on $\mathbf{W}$ only through $\Delta\mathbf{W}$. Therefore, the gradients satisfy $\nabla_{\mathbf{W}}\mathcal{L} = \nabla_{\Delta\mathbf{W}}\mathcal{L}$, because $\partial\Delta\mathbf{W}/\partial\mathbf{W} = \mathbf{I}$. Let $\mathbf{F}_{\mathbf{W}}$ denote the empirical Fisher Information Matrix estimated over weight matrix $\mathbf{W}$:

$$\mathbf{F}_{\mathbf{W}} = \mathbb{E}_{x \sim \mathcal{D}} \left[ \text{vec}(\nabla_{\mathbf{W}}\mathcal{L}) \, \text{vec}(\nabla_{\mathbf{W}}\mathcal{L})^{\top} \right]$$

Using the gradient equivalence above, the Fisher matrix in the $\Delta\mathbf{W}$ space is identical:

$$\mathbf{F}_{\Delta\mathbf{W}} = \mathbb{E}_{x \sim \mathcal{D}} \left[ \text{vec}(\nabla_{\Delta\mathbf{W}}\mathcal{L}) \, \text{vec}(\nabla_{\Delta\mathbf{W}}\mathcal{L})^{\top} \right] = \mathbf{F}_{\mathbf{W}}$$

For notational simplicity, we omit the $\text{vec}(\cdot)$ operator and treat $\mathbf{W}$ as a vectorized parameter. The Fisher regularization term can be defined as:

$$\mathcal{R}_{\mathbf{W}} = \frac{1}{2}(\mathbf{W} - \mathbf{W}^*)^{\top} \mathbf{F}_{\mathbf{W}}(\mathbf{W} - \mathbf{W}^*) = \frac{1}{2}\Delta\mathbf{W}^{\top} \mathbf{F}_{\Delta\mathbf{W}}\Delta\mathbf{W}$$

The gradients of $\mathcal{R}_{\mathbf{W}}$ with respect to $\mathbf{A}$ and $\mathbf{B}$ are:

$$\nabla_{\mathbf{A}}\mathcal{R}_{\mathbf{W}} = \nabla_{\Delta\mathbf{W}}\mathcal{R}_{\mathbf{W}} \cdot \mathbf{B}^{\top} = \mathbf{F}_{\Delta\mathbf{W}}\Delta\mathbf{W} \cdot \mathbf{B}^{\top}$$
$$\nabla_{\mathbf{B}}\mathcal{R}_{\mathbf{W}} = \mathbf{A}^{\top} \cdot \nabla_{\Delta\mathbf{W}}\mathcal{R}_{\mathbf{W}} = \mathbf{A}^{\top} \cdot \mathbf{F}_{\Delta\mathbf{W}}\Delta\mathbf{W}$$

with:

$$\nabla_{\Delta\mathbf{W}}\mathcal{R}_{\mathbf{W}} = \mathbf{F}_{\Delta\mathbf{W}}\Delta\mathbf{W}$$

Therefore, by propagating the gradient through the low-rank decomposition $\Delta\mathbf{W} = \mathbf{AB}$, the Fisher regularization over $\Delta\mathbf{W}$ imposes a constraint on the update directions of the low-rank factors.

### A.2 Implementation Details

This section provides additional details on the method described in Section 3 and the experimental setup in Section 4 of the main text. In particular, we present the algorithm of EWC-LoRA and discuss the effects of hyperparameters, accompanied by additional ablation studies.

### A.2.1 Optimization Algorithms

Algorithm 1 outlines the learning procedure of EWC-LoRA across tasks $\mathcal{T}_1$ to $\mathcal{T}_T$. At each task, the model is updated via low-rank adaptation while incorporating EWC regularization computed from the Fisher Information Matrix. Unlike prior methods that assign task-specific modules, EWC-LoRA regularizes a shared LoRA, thereby maintaining constant memory cost. For clarity, the key equation referenced in the main text is restated here.

$$\mathcal{L}_t'(\mathbf{A}, \mathbf{B}) = \mathcal{L}_t(\mathbf{A}, \mathbf{B}) + \frac{\lambda}{2} \text{vec}(\mathbf{AB})^{\top} \mathbf{F}_{t-1}^{\text{cum}} \text{vec}(\mathbf{AB}) \tag{3}$$

where $\text{vec}(\mathbf{AB})$ is flattened $\mathbf{AB}$. $\mathbf{F}_t^{\text{cum}}$ denotes the accumulated Fisher matrix obtained from task $\mathcal{T}_{t-1}$. $\lambda$ is the regularization strength.

---

**Algorithm 1:** The learning procedure of EWC-LoRA.

---

**Input:** A sequence of tasks $\{\mathcal{T}_t\}_{t=1}^T$ with datasets $\{\mathcal{D}_t\}_{t=1}^T$; A frozen pre-trained model with parameters $\mathbf{W}_0 \in \mathbb{R}^{d_O \times d_I}$; Low-rank adaptation matrices $\mathbf{A} \in \mathbb{R}^{d_O \times r}$ and $\mathbf{B} \in \mathbb{R}^{r \times d_I}$, with rank $r$; Task decay factor $\gamma_t = 0.9,\ \forall t = 1, \ldots, T$.
**Output:** Adapted model parameters $\mathbf{W}_T \in \mathbb{R}^{d_O \times d_I}$ that generalize across all tasks $\{\mathcal{T}_t\}_{t=1}^T$.

Initialize accumulated Fisher Information Matrix: $\mathbf{F}_0^{\mathrm{cum}} \leftarrow \gamma_{\mathrm{prior}} \cdot \mathbf{I}$ ;   `// 0 if no prior`

**for** $t = 1$ **to** $T$ **do**

    **Step 1:** Initialize $\mathbf{A} \leftarrow \mathbf{0}$, $\mathbf{B} \leftarrow \mathcal{U}(a, b)$

    **Step 2:** Low-rank adaptation on current task $\mathcal{T}_t$: $\mathbf{A}^*, \mathbf{B}^* \leftarrow$ Eq. 3 ;

    **Step 3:** Fisher estimation for $\mathcal{T}_t$;

        1. Get gradient $\nabla_\mathbf{W} \mathcal{L}$ on the full-dimensional space ;

        2. Compute diagonal entries of Fisher matrix: $F_t^{i,i} \approx \frac{1}{|\mathcal{D}_t|} \sum_{(x,y) \in \mathcal{D}_t} \left( \frac{\partial \log p_{\mathbf{W}_t^*}(y|x)}{\partial w_i} \right)^2$ ;

        3. Update accumulated Fisher Information Matrix: $\mathbf{F}_t^{\mathrm{cum}} \leftarrow \gamma_t \cdot \mathbf{F}_{t-1}^{\mathrm{cum}} + \mathbf{F}_t$;

    **Step 4:** Integrate low-rank parameter: $\mathbf{W}_t = \mathbf{W}_{t-1} + \mathbf{A}^*\mathbf{B}^*$;

    **Step 5:** Discard task-specific dataset $\mathcal{D}_t$ and Fisher matrix $\mathbf{F}_t$.

---

### A.2.2 HYPERPARAMETERS

We follow the training configurations specified by the authors in the original papers. When such configurations are not available, we adopt a unified set of hyperparameters that has been validated to perform reliably across methods, thereby ensuring consistency in our experimental setup. In all experiments, we use the Adam optimizer with $\beta_1 = 0.9$ and $\beta_2 = 0.99$. The number of training epochs depends on the specific dataset, and the training batch size is set to 128. No shuffling is applied during training; however, we verified that enabling shuffling generally leads to improved results. Table 8 summarizes the hyperparameters used during training, with method-specific parameters highlighted for each respective approach.

Table 8: Hyperparameters for different benchmarks and methods. "**lr**" denotes learning rate. For the parameter $\epsilon$, we refer readers to Liang & Li (2024) for details.

| Datasets | Methods |
|---|---|
| CIFAR-100 | **optimizer:** Adam; **schedular:** Cosine; **batch size:** 128; **shuffle:** False; **epochs:** 20; **rank:** 10 
 **lr:** 0.0005; **classifier lr:** 0.005; **lr decay:** 0.1 
 **lr:** 0.008; **classifier lr:** 0.008; **lr decay:** 0.1 (SD-LoRA) 
 $\epsilon$ : 0.95 (InfLoRA) |
| DomainNet | **optimizer:** Adam; **schedular:** Cosine; **batch size:** 128; **shuffle:** False; **epochs:** 5; **rank:** 30 
 **lr:** 0.0005; **classifier lr:** 0.005; **lr decay:** 0.1 
 **lr:** 0.02; **classifier lr:** 0.02; **lr decay:** 0.0 (SD-LoRA) 
 $\epsilon$ : 0.95 (InfLoRA) |
| ImageNet-R | **optimizer:** Adam; **schedular:** Cosine; **batch size:** 128; **shuffle:** False; **epochs:** 50; **rank:** 10 
 **lr:** 0.0005; **classifier lr:** 0.0005; **lr decay:** 0.1 
 **lr:** 0.01; **classifier lr:** 0.01; **lr decay:** 0.0; **weight decay:** 0.0005 (SD-LoRA) 
 **weight decay:** 0.005 (EWC-LoRA) 
 $\epsilon$ : 0.98 (InfLoRA) |
| ImageNet-A | **optimizer:** Adam; **schedular:** Cosine; **batch size:** 128; **shuffle:** False; **epochs:** 10; **rank:** 10 
 **lr:** 0.0005; **classifier lr:** 0.005; **lr decay:** 0.1 
 $\epsilon$ : 0.98 (InfLoRA) |

For the regularization strength $\lambda$ in Eq. 3, we evaluated a wide range from $10^1$ to $10^{10}$. The trend is illustrated in Figure 3b of the main text. We observe that across all datasets, when using the empirical Fisher, the best overall performance is achieved around $10^7$. Moreover, with relatively

smaller values of $\lambda$ (e.g., $10^1$ to $10^4$), performance tends to decrease initially. For all experiments, the regularization strength $\lambda$ is set to $10^7$.

For the parameter $\gamma$ in Algorithm 1, we set all tasks to be equally important to $\gamma = 0.9$. To further assess its effect, we evaluate the effect of varying $\gamma$ using a single seed for performance comparison. The results are presented in Table 9 and Figure 4. The parameter $\gamma$ controls the accumulation of Fisher information across tasks in continual learning. When $\gamma = 0$, no Fisher information is carried over from previous tasks, meaning that only the Fisher matrix of the current task is used to regularize the subsequent task. As $\gamma$ increases, past information is accumulated more strongly, which can help preserve knowledge from earlier tasks.

As shown in Table 9, the final accuracy $\overline{A}$ remains relatively similar across different $\gamma$ settings. However, notable differences are observed in terms of stability and plasticity. From Figure 4, the trends associated with different $\gamma$ values are clearly visible. We find that there exists a broad range of $\gamma$ values that achieve a similar stability-plasticity trade-off. As indicated by the green-shaded region, the method is not overly sensitive to the exact choice of $\gamma$.

Table 9: Performance comparison under different values of $\gamma$ on CIFAR-100.

|  | $\gamma = 1.0$ | $\gamma = 0.9$ | $\gamma = 0.7$ | $\gamma = 0.5$ | $\gamma = 0.3$ | $\gamma = 0$ |
|---|---|---|---|---|---|---|
| $\overline{A}_{10}$ | 87.64 | 87.47 | 88.07 | 88.14 | **88.22** | 86.63 |
| Avg. | 91.96 | 92.12 | 92.42 | 92.34 | **92.46** | 92.04 |
| Stability | 94.46 | 94.45 | 94.28 | 94.37 | **94.51** | 91.81 |
| Plasticity | 97.66 | 97.99 | 98.35 | 98.33 | 98.29 | **99.07** |

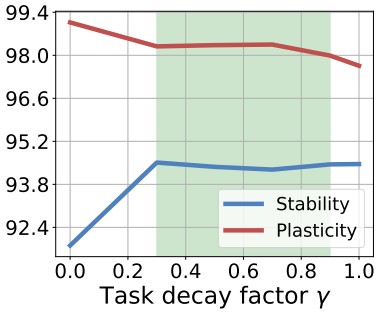

Figure 4: Stability-Plasticity trade-off with various task decay factor $\gamma$. A broad range of $\gamma$ (0.3-0.9) yields a similar trade-off, indicating that the method is not sensitive to the precise choice of $\gamma$.

We further illustrate the effect of $\gamma$ using a per-task accuracy matrix. In Figure 5, each row represents the performance on all previously encountered tasks (x-axis) after learning the current task (y-axis). The results show that setting $\gamma$ to zero causes a significant drop in stability, with earlier tasks experiencing more severe forgetting. In contrast, higher $\gamma$ values help preserve performance on previous tasks while still enabling effective learning of new tasks.

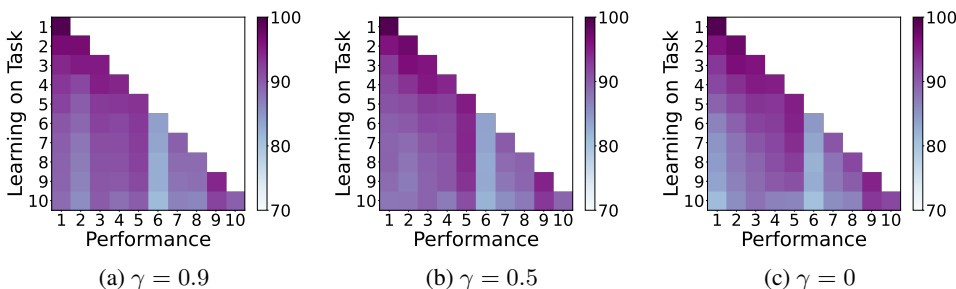

(a) $\gamma = 0.9$      (b) $\gamma = 0.5$      (c) $\gamma = 0$

Figure 5: Task-wise performance on CIFAR-100 under different $\gamma$ settings.

Table 10: Task order of the standard CL benchmark for language tasks.

|  | Task 1 | Task 2 | Task 3 | Task 4 |
|---|---|---|---|---|
| Order 1 | dbpedia | amazon | yahoo | ag |
| Order 2 | dbpedia | amazon | ag | yahoo |
| Order 3 | yahoo | amazon | ag | dbpedia |

## A.3 ADDITIONAL EXPERIMENTS

### A.3.1 COMPARISON RESULTS

**Results on standard CL benchmark for language tasks.** To further evaluate the applicability of EWC-LoRA beyond vision tasks, we extend our experiments to natural language processing scenarios using the T5-large and LLaMA-3.2-1B-Instruct pretrained models. Following the standard language CL benchmark and previous work (Wang et al., 2023), we use three task orders composed of four text classification datasets, as summarized in Table 10. We compare EWC-LoRA with O-LoRA and TreeLoRA under identical training settings. The task-wise performance is evaluated using average accuracy, and the results are shown in Figures 6 and 7. For the larger model T5-large, the effectiveness of EWC-LoRA is more pronounced, as it mitigates forgetting on most tasks across all three orders. For the smaller model LLaMA-3.2-1B-Instruct, EWC-LoRA also exhibits less forgetting compared with the other two methods.

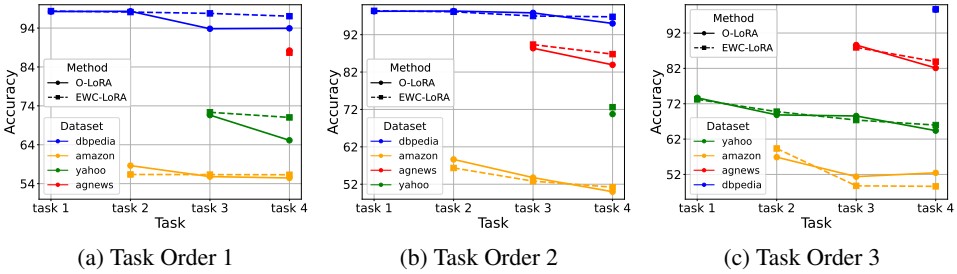

(a) Task Order 1      (b) Task Order 2      (c) Task Order 3

Figure 6: Task-wise performance across the three task orders using the T5-large model.

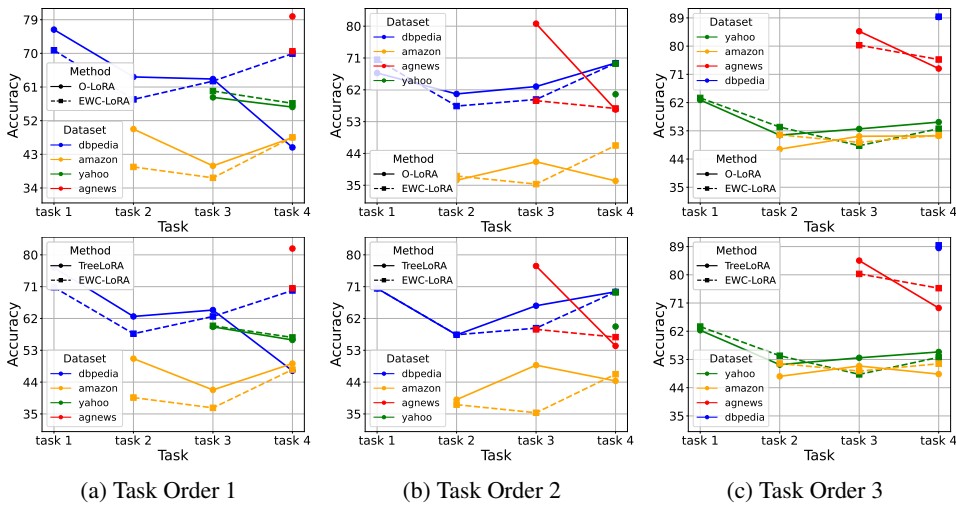

(a) Task Order 1      (b) Task Order 2      (c) Task Order 3

Figure 7: Task-wise performance across the three task orders using the LLaMA-3.2-1B-Instruct model.

**Results Across Varied Task Length.** Table 11 reports the full results of low-rank continual learning methods under varying task lengths on ImageNet-R. For EWC-LoRA, the performance gains

appear relatively modest, likely due to the inherent domain shift in this benchmark, which may constrain the effectiveness of regularization-based methods for continual adaptation. We further observe that InfLoRA achieves better results on shorter sequences, whereas SD-LoRA performs more strongly on longer sequences. Examining the accuracy matrix for longer sequences, we find that although SD-LoRA exhibits lower stability, its higher plasticity often leads to superior final performance. This observation suggests that model evaluation should go beyond reporting only the final accuracy. It is also important to track performance throughout the task sequence and explicitly report both stability and plasticity.

Table 11: Comparison results on ImageNet-R across different task lengths (in %).

| Tasks | **ImageNet-R** (N=5) | | **ImageNet-R** (N=20) | |
|---|---|---|---|---|
| Methods | $\overline{A}_5$ ($\uparrow$) | Avg. ($\uparrow$) | $\overline{A}_{20}$ ($\uparrow$) | Avg. ($\uparrow$) |
| Joint Train | $81.69_{(0.30)}$ | $85.57_{(0.13)}$ | $81.66_{(0.22)}$ | $86.41_{(0.10)}$ |
| Finetune | $69.26_{(0.74)}$ | $78.88_{(0.31)}$ | $47.06_{(2.05)}$ | $63.01_{(0.56)}$ |
| InfLoRA | $\mathbf{77.37}_{(0.30)}$ | $\mathbf{82.19}_{(0.24)}$ | $69.63_{(0.62)}$ | $76.95_{(0.54)}$ |
| SD-LoRA | $74.90_{(1.58)}$ | $79.93_{(0.29)}$ | $\mathbf{72.26}_{(0.37)}$ | $\mathbf{77.81}_{(0.21)}$ |
| Vanilla LoRA | $70.15_{(1.00)}$ | $79.16_{(0.37)}$ | $56.17_{(1.50)}$ | $69.51_{(0.37)}$ |
| EWC-LoRA | $\underline{76.36}_{(0.21)}$ | $\underline{81.43}_{(0.13)}$ | $\underline{70.18}_{(1.06)}$ | $\underline{77.06}_{(0.54)}$ |

**Task-wise Performance.** Figure 8 and Figure 9 show the accuracy matrix of the LoRA-based methods on CIFAR-100 and DomainNet. Each row corresponds to the performance on all previously encountered tasks after training on the current task. The diagonal entries correspond to the most recently trained tasks. As expected, Finetune exhibits severe forgetting on previous tasks while maintaining high performance on the current task, as shown in the matrix entries. On CIFAR-100, we observe that InfLoRA better preserves performance on the earliest task (first column). On DomainNet, SD-LoRA adapts more effectively to new tasks. On both datasets, EWC-LoRA achieves a more balanced trade-off between stability and plasticity compared to the other two methods.

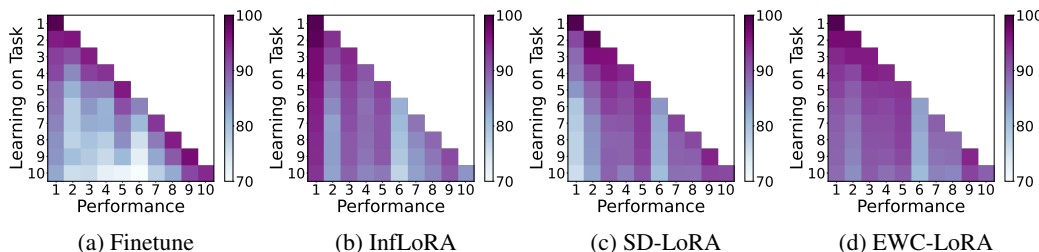

   (a) Finetune       (b) InfLoRA       (c) SD-LoRA       (d) EWC-LoRA

Figure 8: Task-wise performance of LoRA-based methods on CIFAR-100. Each row represents the performance on all previously encountered tasks (x-axis) after learning the current task (y-axis).

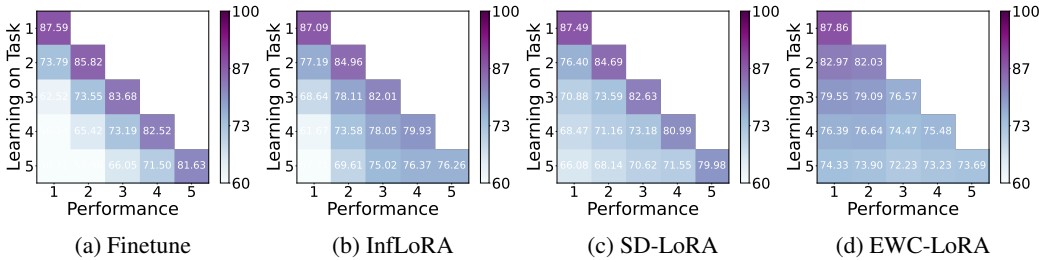

   (a) Finetune       (b) InfLoRA       (c) SD-LoRA       (d) EWC-LoRA

Figure 9: Task-wise performance of LoRA-based methods on DomainNet. Each row represents the performance on all previously encountered tasks (x-axis) after learning the current task (y-axis).

**Trade-off between Stability and Plasticity.**    To better understand how different methods balance stability and plasticity, we introduce a trade-off metric that approximates this balance:

$$\mathrm{T} = \frac{2 \cdot S \cdot P}{S + P} \tag{8}$$

where $S$ and $P$ denote the Stability and Plasticity, respectively, as defined in Eq. 5 and Eq. 6. Figure 10 illustrates the trade-off between stability and plasticity for different low-rank CL methods. Vanilla LoRA generally achieves the highest plasticity, as it does not include mechanisms to prevent forgetting. EWC-LoRA attains stability comparable to InfLoRA, while retaining more plasticity than InfLoRA. Overall, EWC-LoRA achieves the best trade-off among the methods.

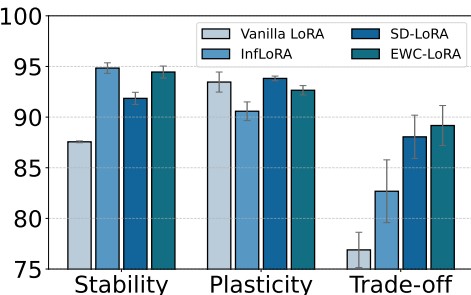

Figure 10: Trade-off between stability and plasticity.

**Ablation on LoRA Ranks.**    In our main experiments, we used a fixed LoRA rank across all tasks to ensure a fair comparison. Here, we explore whether more complex datasets benefit from different rank settings. Specifically, we conducted an ablation study with different rank ($r$) values on two benchmarks of varying difficulty, and the results are summarized in 12. We find that increasing or decreasing the rank $r$ does lead to some variation in performance. However, the difference is relatively small, which indicates that the effective parameter space of the model is inherently low-rank, and that the regularization applied within this low-rank space follows the same principle.

Table 12: Ablation study on LoRA ranks $r$ across two benchmarks of varying difficulty.

| | CIFAR-100 | | ImageNet-R | |
|---|---|---|---|---|
| $r$ | $\overline{A}_{10}$ | Avg. | $\overline{A}_{10}$ | Avg. |
| 16 | 88.31 | 92.59 | 72.10 | 79.18 |
| 10 | 87.91 | 92.27 | 72.86 | 78.95 |
| 4 | 87.79 | 92.12 | 72.42 | 78.78 |
| 1 | 86.77 | 91.61 | 70.37 | 77.27 |

To exclude the effect of additional regularization, we examine the stability-plasticity trade-off using vanilla LoRA. We observe that a vanilla LoRA with a lower rank (e.g., $r = 1$) even outperforms higher-rank settings in terms of final average accuracy. The corresponding results are presented in Table 13. The reason is that a smaller rank naturally provides the model with stronger stability, but at the cost of reduced plasticity. For CIFAR-100, this cost of plasticity is relatively minor, whereas for ImageNet-R, it becomes much more pronounced. This indicates that for more challenging benchmarks, the model should adopt a larger rank to ensure sufficient plasticity.

### A.3.2   FISHER ESTIMATION ANALYSIS

**Precomputed Fisher.**    As illustrated in Xiang et al. (2023) and Šliogeris et al. (2025), they use a precomputed Fisher Information Matrix to preserve prior knowledge. Following this approach, we evaluate performance when a Fisher Information Matrix is precomputed and used throughout the continual learning process. Unlike these works, we do not rely on a large-scale dataset to compute the Fisher matrix. We consider two settings: (1) Uniform parameter importance: All parameters are assigned equal importance, i.e., $\gamma_{\mathrm{prior}}$ in Algorithm 1 is set to a constant, and the Fisher matrix is an

Table 13: Ablation study on LoRA ranks ($r$) in the Vanilla LoRA setting for CIFAR-100 and ImageNet-R. Note that the plasticity is computed based on the upper bound at rank 10. When the rank is 16, the plasticity can exceed 100%.

| | CIFAR-100 | | | | ImageNet-R | | | |
|---|---|---|---|---|---|---|---|---|
| $r$ | $\overline{A}_{10}$ | Avg. | Stability | Plasticity | $\overline{A}_{10}$ | Avg. | Stability | Plasticity |
| 16 | 83.05 | 89.71 | 86.25 | 100.42 | 66.05 | 76.26 | 74.17 | 101.41 |
| 10 | 82.99 | 89.74 | 87.56 | 98.86 | 66.32 | 76.35 | 75.69 | 99.86 |
| 4 | 83.35 | 90.10 | 88.70 | 98.24 | 67.52 | 76.33 | 77.27 | 97.30 |
| 1 | 83.72 | 89.94 | 89.95 | 97.36 | 68.43 | 77.09 | 81.95 | 96.75 |

identity matrix. (2) Dataset-based Fisher: The Fisher Information Matrix is computed in advance using the entire dataset. The results are shown in Table 14. We observe that the uniform Fisher exhibits lower stability, while the plasticity of both methods is similar, but still much lower than ours. This suggests that using a precomputed, fixed Fisher imposes stronger constraints on the weights, thereby limiting plasticity. As expected, the dataset-based Fisher achieves higher stability than the uniform Fisher, which is reasonable since it better captures the true importance of the parameters.

Table 14: Regularization using precomputed Fisher on CIFAR-100.

| Strategy | $\overline{A}_{10}$ | Avg. | Stability | Plasticity |
|---|---|---|---|---|
| Uniform $\mathbf{F} = \mathbf{I}$ | 83.02 | 88.85 | 92.26 | 94.63 |
| Dataset-based $\mathbf{F}$ | 83.87 | 89.36 | 93.15 | 94.74 |

**Computation on Fisher Information Matrix.** As suggested by van de Ven (2025), we investigate different methods for estimating the FIM. The results are presented in Table 15. Here, "Exact" indicates that the inner expectation in Eq. 4 is computed exactly for each training sample. "Exact (n=500)" denotes that the outer expectation is calculated using a subset of 500 samples from the old training data. "Sample" indicates that the inner expectation is computed over a sampled class. The results indicate that the optimal regularization strength varies according to the estimation method. In general, the Exact Fisher outperforms the Empirical Fisher, requiring a smaller regularization strength. The Sample method yields slightly better results than the Empirical Fisher.

Table 15: Different ways for estimating the Fisher matrix. Final accuracy of each variant using its optimal strength $\lambda$ on CIFAR-100.

| Estimation | $\overline{A}_{10}$ | Avg. | Best $\lambda$ |
|---|---|---|---|
| Exact | 88.32 | 92.77 | $\lambda = 10^5$ |
| Exact (n=500) | 88.28 | 92.76 | $\lambda = 10^5$ |
| Sample | 88.10 | 92.50 | $\lambda = 10^7$ |
| Empirical | 87.91 | 92.27 | $\lambda = 10^7$ |

**Accuracy of the Fisher Estimation.** In rehearsal-free EWC, the Fisher matrix for the current task is computed only once at the end of training and is not updated thereafter, which may result in a stale Fisher estimate when the parameters drift substantially from the task-specific optimum (Wu et al., 2024). We further investigate whether a similar issue exists in low-rank CL methods. To this end, we track the evolution of the Fisher matrix for each task throughout the learning process. Specifically, for a given task $i$, we first compute its Fisher matrix $\mathbf{F}_i^{(i)}$. The model is then sequentially trained on subsequent tasks $i+1, i+2, \ldots, t$. We evaluate the accuracy of the Fisher Information Matrix from two perspectives: **(1) Scale sensitivity**, which indicates whether the parameter importance encoded by the Fisher matrix becomes degraded or inflated. **(2) Structural pattern**, which reflects whether the set of parameters that EWC aims to protect changes over time.

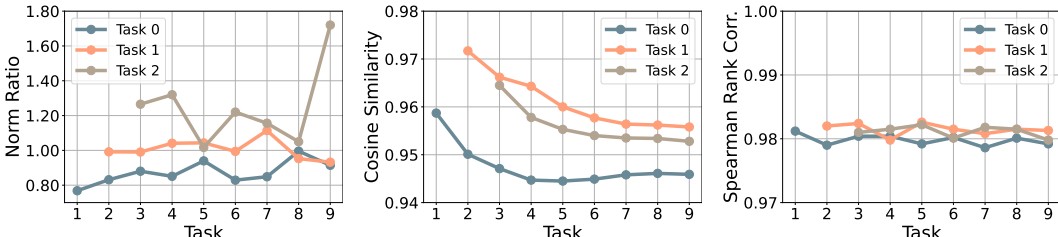

Figure 11: Evaluation of Fisher accuracy as the task index increases on CIFAR-100 with 10 tasks.

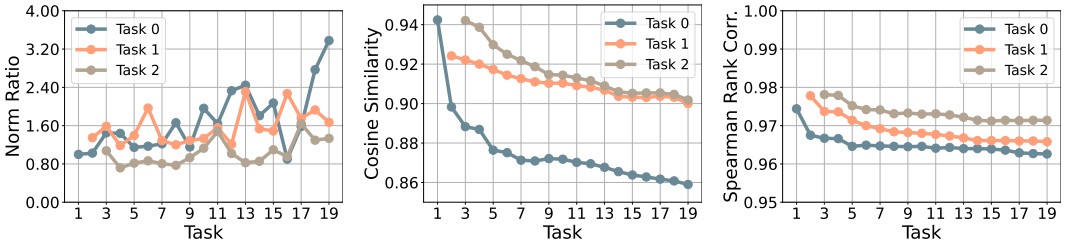

Figure 12: Evaluation of Fisher accuracy as the task index increases on CIFAR-100 with 20 tasks.

To evaluate scale sensitivity, we recompute the Fisher matrices $\mathbf{F}_t^{(i)}$ using the current model (subscript $t$) and the data from task $i$ (superscript $(i)$) after learning each new task. We use the Norm Ratio to quantify the numerical changes in the Fisher matrix. The Norm Ratio is defined as:

$$NR = \frac{||\mathbf{F}_t^{(i)}||}{||\mathbf{F}_i^{(i)}||} \tag{9}$$

where $NR > 1$ indicates that the Fisher matrix has become inflated, and $NR < 1$ indicates that it has degraded.

To quantify changes in the structural patterns of the Fisher matrix, we measure the similarity between the Fisher matrix estimated at task $t$ and the original Fisher matrix. We consider two settings: (1) Rehearsal-free: we compare the accumulated Fisher matrix $\mathbf{F}_t^{\mathrm{cum}}$ with the original optimal Fisher matrix $\mathbf{F}_i^{(i)}$, and (2) Rehearsal-based: we compute the optimal Fisher matrix $\mathbf{F}_t^{(1...t)}$ for the current task by incorporating data from previous tasks, and compare $\mathbf{F}_t^{(1...t)}$ with $\mathbf{F}_i^{(i)}$. We employ Spearman Rank Correlation to capture the overall consistency and Cosine Similarity to assess shifts in the most critical task parameters. The Spearman Rank Correlation evaluates the agreement in rank ordering between two vectors. Given two vectors $\mathbf{v}_1$ and $\mathbf{v}_2$, each element is first converted to its rank, denoted as $r_{1,i} = \mathrm{rank}(v_{1,i})$ and $r_{2,i} = \mathrm{rank}(v_{2,i})$. The Spearman correlation coefficient is then computed as:

$$\rho = 1 - \frac{6 \sum_{i=1}^n (r_{1,i} - r_{2,i})^2}{n(n^2 - 1)} \tag{10}$$

where $n$ represents the number of elements in the vectors. The Spearman correlation coefficient $\rho \in [-1, 1]$, where $\rho = 1$ indicates identical rank ordering, 0 indicates no correlation, and -1 indicates complete inverse ordering. Spearman correlation captures the overall pattern similarity between two vectors regardless of their scale.

Cosine Similarity focuses on the directional alignment between two vectors. A high cosine similarity indicates that the important parameters in one model are largely aligned with those in the other, capturing the consistency of parameter sensitivity patterns across models. The cosine similarity between $\mathbf{F}_t^{\mathrm{cum}}$ and $\mathbf{F}_i^{(i)}$ is defined as:

$$\mathrm{CosSim}(\mathbf{F}_t^{\mathrm{cum}}, \mathbf{F}_i^{(i)}) = \frac{\mathbf{F}_t^{\mathrm{cum}} \cdot \mathbf{F}_i^{(i)}}{||\mathbf{F}_t^{\mathrm{cum}}|| \, ||\mathbf{F}_i^{(i)}||} \tag{11}$$

We report the above metrics for the three oldest tasks throughout the learning process, and the results are shown in Figure 11, 12 and 13. We observe that the Norm Ratio fluctuates around 1, indicating

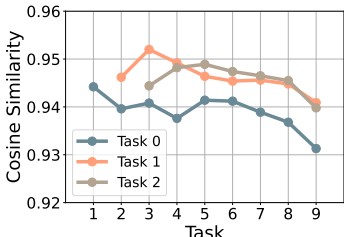 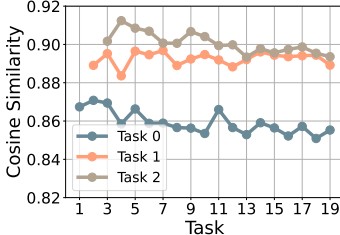

Figure 13: Cosine similarity between two optimal Fisher matrices under the rehearsal-based setting.

that the strength of the constraint imposed on old tasks changes as the model learns new ones. As for the structural pattern of the Fisher Information Matrix, we observe that as the model learns more tasks, the Cosine Similarity gradually decreases, whereas the Spearman Rank Correlation shows a much milder decline. This indicates that although the absolute directions of parameter importance shift with new tasks, the relative ordering of important parameters remains largely stable.

Wu et al. (2024) has demonstrated the staleness of Fisher estimates in full-parameter EWC. In low-rank continual learning, a similar issue also exists for EWC. Under the rehearsal-based setting, however, the Fisher matrix remains relatively stable compared to the rehearsal-free setting. As the model learns more tasks, the key parameters identified by the Fisher matrix remain largely consistent, and the observed inaccuracies arise primarily from changes in magnitude and directional scaling, rather than from a fundamental reordering of parameter importance.

## A.4 THE USE OF LARGE LANGUAGE MODELS (LLMS)

In the preparation of this manuscript, the LLM was used to refine sentence structures, ensure clarity, and improve the readability of the text.

