# OpenReview forum: "Revisiting Weight Regularization for Low-Rank Continual Learning"
_ICLR.cc/2026/Conference — ICLR 2026 Poster_

### Official Review · Reviewer_gUkm · 2025-10-26

**Soundness:** 3
**Presentation:** 3
**Contribution:** 3
**Rating:** 6
**Confidence:** 4

**Summary:**

This manuscript focused on the problem of continual learning with pre-trained model (CL-PTM), specifically, parameter-efficient continual learning with low-rank adapters. To mitigate the task inferences, this manuscript revisited the weight regularization method, which was less explored under the context of low-rank continual learning. Instead of applying task-specific LoRA modules, this manuscript proposed to adopt a shared low-rank update to keep the storage requirement constant while applying the regularization on the updated parameters with EWC. Specifically, the parameter importance in EWC was estimated over the full-dimensional space. The authors conducted experiments on the proposed method EWC-LoRA, showing that the proposed method achieved state-of-the-art performance compared to existing methods without sacrificing efficiency.

**Strengths:**

1. Compared to previous methods with task-independent LoRA modules, the proposed EWC-LoRA continuously merged the task-specific parameters into the previous ones. The model was trained in a parameter-efficient manner, and the maintained parameters for all previous tasks remained constant during the inference after weight merging.
2. The proposed EWC-LoRA adopted a dynamic way to update the Fisher Information Matrix for EWC, avoiding the fixed precomputed FIM.
3. The ablation studies were relatively complete.

**Weaknesses:**

1. The proposed approach is mainly an adaptation of EWC to LoRA; the innovation lies in its parameterization and efficiency rather than a fundamentally new learning principle.
2. The update of the accumulated FIM seems heuristic. Although the authors provided some discussions about the choices of $\gamma$ during accumulated FIM update, it seemed that the choice of this hyperparameter was still not principled.
3. Some recent LoRA-based PECL methods were not involved in the experimental comparisons.

**Questions:**

1. Regarding the training time shown in Table 3. It seems that the per-task training time seemed almost the same as the vanilla LoRA method. However, it seems that Eq. (4) needs to compute the gradient over the full dataset. I wonder if this procedure took such little time considering it requires the backpropagation on the whole dataset.
2. The LoRA-based baselines considered in this manuscript were InfLoRA and O-LoRA. However, recently, many studies about PECL have been published, even considering only the interference reduction in LoRA-based PECL. Could the author provide more comparisons with the methods in this stream, including but not limited to [1,2].

References:

[1] CL-LoRA: continual low-rank adaptation for rehearsal-free class-incremental learning. CVPR 2025
[2] BiLoRA: almost-orthogonal parameter spaces for continual learning. CVPR 2025.

---

> ### Author Response · Authors · 2025-11-21
> **Response to Reviewer gUkm (Weakness 1, 2)**
>
> We thank the Reviewer for the time spent on our work and the suggestions for improving its quality. Below, we address the points raised by the Reviewer, and we have made the corresponding revisions to the main text, which has greatly improved the quality of our work.
>
> ## (Weakness 1) The contribution of our work.
>
> We thank the reviewer for pointing this out. We agree that our work builds upon the principles of practical regularization-based low-rank continual learning (CL) methods. We would like to emphasize that directly applying EWC to large-scale models can be computationally prohibitive or lead to suboptimal performance. Our paper presents the first systematic investigation of regularization-based low-rank CL methods and provides insights and conclusions that we hope can guide future research in this direction.
>
> ## (Weakness 2) The update of the accumulated Fisher Information Matrix (FIM).
>
> We thank the reviewer for raising this concern. We acknowledge that the choice of the task decay factor $\gamma$ in our experiments is heuristic. To address this point, we surveyed recent literature on EWC-related methods [2,3] and found that most existing works rely on empirically chosen values or simply follow the settings used in Online-EWC [1] (e.g., $\gamma=1$ or $\gamma=0.9$).
>
> We further conducted additional experiments under the low-rank adaptation framework on a wide range of $\gamma$. As shown in **Table 1**, the final accuracy $\overline{A}$ remains relatively similar across different $\gamma$ settings. However, notable differences are observed in terms of stability and plasticity. We include **Figure 4** in the Appendix to better illustrate the effect of different values of $\gamma$. We find that there exists a broad range of $\gamma$ values (0.3~0.9) that achieve a similar stability–plasticity trade-off. As indicated by the green-shaded region, the method is not overly sensitive to the exact choice of $\gamma$.
>
> **Table 1.** Performance comparison under different values of $\gamma$ on CIFAR-100
> ㅤ | $\gamma=1.0$ | $\gamma=0.9$ | $\gamma=0.7$ | $\gamma=0.5$ | $\gamma=0.3$ | $\gamma=0$
> -----|-----|-----|-----|-----|-----|-----
> $\overline{A}_{10}$ | 87.64 | 87.47 | 88.07 | 88.14 | **88.22** | 86.63
> Avg. | 91.96 | 92.12 | 92.42 | 92.34 | **92.46** | 92.04
> Stability | 94.46 | 94.45 | 94.28 | 94.37 | **94.51** | 91.81
> Plasticity | 97.66 | 97.99 | 98.35 | 98.33 | 98.29 | **99.07**
>
> Although we do not yet have a principled conclusion on the optimal setting of this parameter, we hope that these experiments provide practical insights into the choice of $\gamma$ for regularization-based methods under low-rank adaptation. We also include this discussion in **Appendix A.2.2** (Line 982-1007) to provide a more comprehensive analysis of our work, as suggested by the reviewer.
>
> reference:
>
> [1] Progress & compress: A scalable framework for continual learning, ICML 2018.
>
> [2] On the Computation of the Fisher Information in Continual Learning, ICLR blogpost 2025.
>
> [3] Unbiased Online Curvature Approximation for Regularized Graph Continual Learning, arXiv 2025.

---

> ### Author Response · Authors · 2025-11-21
> **Response to Reviewer gUkm (Weakness 3 and Question 1, 2)**
>
> ## (Weakness 3 and Question 2) Comparison with other LoRA-based PECL methods.
>
> We sincerely thank the reviewer for highlighting the importance of including additional methods from this research line. Following the reviewer’s suggestion, we have expanded the experimental comparisons in **Section 4.2** (Main Results). Specifically, we now incorporate **(1) CL-LoRA and BiLoRA** on the four vision benchmarks and **(2) O-LoRA and Tree-LoRA** on the two natural language benchmarks. All experiments were conducted using the official implementations and recommended hyperparameter settings. The newly added results for the two benchmarks are presented in Table 1 and Table 2.
>
> **Table 1.** Comparison results on CIFAR-100, DomainNet, ImageNet-R, and ImageNet-A (in %).
> Methods | CIFAR-100 |  ㅤ | DomainNet |  ㅤ | ImageNet-R |  ㅤ | ImageNet-A |  ㅤ
> -------|-----------|-----------|-----------|-----------|-----------|-----------|-----------|-----------
>  ㅤ | $\overline{A}_{10}$ | Avg. | $\overline{A}_{5}$ | Avg. | $\overline{A}_{10}$ | Avg. | $\overline{A}_{10}$ | Avg.
> CL-LoRA | 87.65$\pm$0.53 | 92.25$\pm$0.41 | 71.06$\pm$0.31 | 77.76$\pm$0.29 | **78.72**$\pm$0.44 | **85.20**$\pm$0.45 | 57.62$\pm$0.89 | **70.76**$\pm$0.63
> BiLoRA  | 85.99$\pm$0.49 | 90.62$\pm$0.42 | 69.75$\pm$0.23 | 73.86$\pm$0.21 | 74.28$\pm$0.92 | 77.38$\pm$0.88 | 51.05$\pm$0.74 | 62.82$\pm$0.65
> EWC-LoRA | **87.91**$\pm$0.57 | **92.27**$\pm$0.39 | **73.46**$\pm$0.16 | **79.58**$\pm$0.10 | 72.86$\pm$0.79 | 78.95$\pm$0.86 | **59.89**$\pm$0.26 | 68.33$\pm$0.67
>
> **Table 2.** Comparison results on the standard language CL benchmark with the two pretrained models.
> Backbone | Method | Order-1 | Order-2 | Order-3 | Avg.
> -------|-----------|-----------|-----------|-----------|-----------
> T5-large | O-LoRA | 75.69 | 74.92 | **74.40** | 75.01
>  ㅤ | EWC-LoRA | **78.01** | **76.85** | 74.30 | **76.39**
> LLaMa-3.2-1B-Instruct | O-LoRA | 56.96 | 55.74 | 67.32 | 60.01
>  ㅤ | TreeLoRA | 58.54 | 56.96 | 65.42 | 60.30
>  ㅤ | EWC-LoRA | **61.17** | **60.47** | **67.61** | **63.08**
>
> We thank the reviewer again for this helpful suggestion, which has significantly improved the completeness of our experimental evaluation. We have updated the experimental section accordingly and have also expanded the discussion to provide a more complete analysis.
>
> reference:
>
> [1] CL-LoRA: continual low-rank adaptation for rehearsal-free class-incremental learning, CVPR 2025.
>
> [2] BiLoRA: almost-orthogonal parameter spaces for continual learning, CVPR 2025.
>
> [3] O-LoRA: Orthogonal Subspace Learning for Language Model Continual Learning, EMNLP 2023.
>
> [4] TreeLoRA: Efficient Continual Learning via Layer-Wise LoRAs Guided by a Hierarchical Gradient-Similarity Tree, ICML 2025.
>
>
>
> ## (Question 1)  Analysis of the training time in Table 3.
>
> We thank the reviewer for carefully examining our work. The additional computational cost of computing the Empirical Fisher primarily depends on two factors:
>
> **(1) The number of trainable parameters:** In our implementation, the Fisher information matrix is computed only once after completing the training of each task, and this computation takes roughly the time of one additional epoch. Since each task is typically trained for multiple epochs, the extra cost introduced by computing the Fisher matrix is relatively minor compared with the overall training time of a full task.
>
> **(2) The number of samples:** For datasets with very few samples, such as ImageNet-A, the training time for Vanilla LoRA and EWC-LoRA is nearly identical, as each task contains approximately 750 images. In contrast, on larger datasets such as DomainNet, the difference in computational cost becomes more pronounced, since each task contains roughly 120,000 images.

---

### Official Review · Reviewer_VVBD · 2025-11-01

**Soundness:** 3
**Presentation:** 3
**Contribution:** 3
**Rating:** 6
**Confidence:** 4

**Summary:**

The paper introduces EWC-LoRA, a parameter-efficient CL method that combines EWC with a shared LoRA. Unlike prior low-rank CL approaches that allocate task-specific LoRA modules, EWC-LoRA regularizes a single low-rank update in the full-dimensional space using an accumulated diagonal Fisher matrix. This yields constant memory overhead, competitive accuracy, and a tunable stability–plasticity trade-off.

**Strengths:**

- The paper presents an interesting idea of revisiting weight regularization in the low-rank regime, which effectively bridges traditional CL methods and PECL approaches.
- The authors provide a comprehensive comparison of computational cost and parameter efficiency, clearly demonstrating how the proposed method performs relative to other LoRA-based continual learning baselines.
- The proposed approach shows consistent improvement across multiple datasets and experimental setups, indicating good robustness and generalizability.

**Weaknesses:**

- Accuracy of the Hessian estimation. The paper relies on the empirical Fisher information matrix to estimate the importance of weights for each task. However, as discussed in Meta-CL [1], the Fisher matrix used in EWC-based methods tends to become stale and outdated over time, leading to inaccurate importance estimation. It remains unclear whether a similar issue [1] arises in the PECL setting adopted here.
- Effectiveness under longer task sequences. When the number of tasks increases, it is uncertain whether the accumulated Fisher matrix can still provide reliable importance estimation. Indeed, the performance drop observed on ImageNet-R with N = 20 tasks suggests that the method’s stability may degrade as the task length grows. A more detailed analysis or explanation from the authors would help clarify this behavior.

[1] "Meta continual learning revisited: Implicitly Enhancing Online Hessian Approximation via Variance Reduction." ICLR, 2024.

**Questions:**

Overall, I think this paper makes a valuable and insightful exploration by bridging traditional regularization-based CL methods with the PECL framework. However, I still have some concerns, and I hope the authors can address the issues raised in Weaknesses 1 and 2.

---

> ### Author Response · Authors · 2025-11-21
> **Response to Reviewer VVBD (Weakness 1)**
>
> We thank the Reviewer for the time spent on our work and the suggestions for improving its quality. We appreciate the Reviewer’s mention of Meta-CL. We have included this paper in the updated experimental analysis. Below, we address the points raised by the Reviewer.
>
> ## (Weakness 1) Accuracy of the Hessian estimation.
>
> We thank the reviewer for pointing out this insightful concern raised in Meta-CL. Motivated by this, we further investigate whether a similar issue exists in low-rank CL methods. To this end, we track the evolution of the Fisher matrix for each task throughout the learning process.
>
> Specifically, for a given task $i$, we first compute its Fisher matrix $\mathbf{F}_i^{(i)}$. The model is then sequentially trained on subsequent tasks $i+1$, $i+2$, $\dots$, $t$. We evaluate the accuracy of the Fisher Information Matrix from two perspectives: **(1) Scale sensitivity**, which indicates whether the parameter importance encoded by the Fisher matrix becomes degraded or inflated. **(2) Structural pattern**, which reflects whether the set of parameters that EWC aims to protect changes over time.
>
> We use three metrics to evaluate Fisher accuracy: **(1) Norm Ratio**, which quantifies numerical changes in the Fisher matrix; **(2) Spearman Rank Correlation**, which captures the consistency of the parameter-importance ordering; and **(3) Cosine Similarity**, which assesses directional shifts in the most critical task parameters.
>
> Specifically, we recompute the Fisher matrices $\mathbf{F}_{t}^{(i)}$ using the current model (subscript $t$) and the data from task $i$ (superscript $(i)$) after learning each new task. To quantify changes in the structural patterns of the Fisher matrix, we measure the similarity between the Fisher matrix estimated at task $t$ and the original Fisher matrix. We consider two settings: (1) Rehearsal-free: we compare the accumulated Fisher matrix with the original optimal Fisher matrix, and (2) Rehearsal-based, we compute the optimal Fisher matrix for the current task by incorporating data from previous tasks, and compare it with the original optimal Fisher matrix.
>
> We report the above metrics for the first three tasks (Tasks 1, 2, and 3) throughout the learning process, and the results are presented in **Figures 11**, **12**, and **13** in the **Appendix A.3.2**. The details of the measurement procedure are provided in the corresponding sections. For the **scale sensitivity**, we observe that the Norm Ratio fluctuates around 1, indicating that the strength of the constraint imposed on old tasks changes as the model learns new ones. Regarding the **structural pattern**, we find that the Cosine Similarity gradually decreases as the model learns more tasks, whereas the Spearman Rank Correlation exhibits a milder decline. This suggests that while the absolute directions of parameter importance shift as new tasks are introduced, the relative ordering among important parameters remains largely stable.
>
> Our experiments show that, as suggested in Meta-CL, the accumulated Fisher indeed becomes outdated over time. Under the rehearsal-based setting, however, the Fisher matrix remains relatively stable compared to the rehearsal-free setting. As the model learns more tasks, the key parameters identified by the Fisher matrix remain largely consistent, and the observed inaccuracies arise primarily from changes in magnitude and directional scaling, rather than from a fundamental reordering of parameter importance.
>
> We have incorporated a detailed discussion of these findings in the **Appendix A.3.2** to support further research in this direction.

---

> ### Author Response · Authors · 2025-11-21
> **Response to Reviewer VVBD (Weakness 2)**
>
> ## (Weakness 2) Effectiveness under longer task sequences.
>
> We thank the reviewer for this thought-provoking question. We conducted a further analysis of EWC-LoRA’s effectiveness under longer task sequences on CIFAR-100 and ImageNet-R.
>
> As shown in Table 6 of the main text, our results indicate that with a 20-task sequence, EWC-LoRA consistently outperforms other task-isolated LoRA-based methods on CIFAR-100, but performs less favorably on ImageNet-R. We attribute this to two main factors.
>
> (1) First, as the reviewer noted, the accuracy of the Fisher information estimation degrades as the task sequence becomes longer, which reduces the effectiveness of the regularization. Considering that Meta-CL employs rehearsal to mitigate the staleness of Fisher estimation, incorporating a similar approach in our method could help address this issue.
>
> (2) Second, ImageNet-R exhibits larger domain shifts, requiring the model to retain greater capacity to learn new tasks. To demonstrate this, we evaluate the model using two rank configurations to assess its performance. We find that a smaller rank naturally provides the model with stronger stability, but at the cost of reduced plasticity. For CIFAR-100, this cost in plasticity is relatively minor, whereas for ImageNet-R, the cost in plasticity is much more significant. This indicates that for more challenging datasets, integrating the regularization method with the extended LoRA architecture could be a more effective solution.

---

### Official Review · Reviewer_ETzF · 2025-11-01

**Soundness:** 4
**Presentation:** 4
**Contribution:** 4
**Rating:** 8
**Confidence:** 4

**Summary:**

This paper revisits weight regularization in the context of low-rank continual learning (CL), specifically focusing on Elastic Weight Consolidation (EWC) applied to parameter-efficient continual learning (PECL) with pre-trained models. The authors propose EWC-LoRA, a method that regularizes a shared low-rank update using the Fisher Information Matrix (FIM) estimated in the full-dimensional space. This approach avoids the linear growth in memory with the number of tasks by maintaining a constant memory footprint. The authors provide a systematic analysis of EWC in low-rank CL, demonstrate a superior stability-plasticity trade-off, and validate their method across multiple benchmarks (CIFAR-100, DomainNet, ImageNet-R, ImageNet-A), showing an average improvement of 8.92% over vanilla LoRA and competitive or better performance compared to state-of-the-art low-rank CL methods.

**Strengths:**

+ Strong theoretical grounding and extensive experiments across multiple benchmarks.
+ Well-written with clear motivation, method, and results.
+ Provides a memory-efficient, tunable, and high-performing solution for PECL.

**Weaknesses:**

+ The Fisher estimation, though efficient, still introduces non-negligible memory overhead
+ EWC-LoRA is indeed not very original, but the reviewer acknowledged that it is meaningful to make existing methods work in PEFT and explain why.

**Questions:**

+ The paper mentions sensitivity to dataset complexity—could you discuss how to automatically select or adapt the rank for different tasks?
+ The storage overhead of inflora as well as one efficient version of sd-lora is not increased linearly, could you explain your benefits compared with them? Or why your performance can be better?

---

> ### Author Response · Authors · 2025-11-21
> **Response to Reviewer ETzF (Question 1.1)**
>
> We thank the Reviewer for the time spent on our work and for the encouraging comments on our work. Below, we address the points mentioned by the Reviewer.
>
> ## (Question 1) Discussion on the Method’s Sensitivity and the Effectiveness of the LoRA Rank
>
> We thank the reviewer for this insightful comment. Investigating the model’s sensitivity to the target dataset is indeed a promising direction in regularization-based low-rank continual learning (CL) methods.
>
> - Regarding the selection of the optimal rank for different datasets, we consider this an open question, as it involves a trade-off between performance and computational cost. Nevertheless, we conducted a series of ablation experiments on the LoRA rank $r$ to explore whether more complex datasets benefit from different rank settings (Q1.1).
>
> - Additionally, we provide a further discussion on why the regularization-based low-rank CL method, EWC-LoRA, performs slightly worse than other LoRA-based counterparts on the ImageNet-R benchmark, highlighting a possible direction for improvement (Q1.2).
>
> ### (Q1.1) Ablation study on different LoRA Ranks ($r$) across two benchmarks.
>
> In our original experiments, we used a fixed LoRA rank across all tasks to ensure a fair comparison. As suggested by the reviewer, we conducted an ablation study with different rank ($r$) values on two benchmarks of varying difficulty, and the results are summarized in **Table 1**. We find that increasing or decreasing the rank $r$ does lead to some variation in performance. However, the difference is relatively small, which indicates that the effective parameter space of the model is inherently low-rank, and that the regularization applied within this low-rank space follows the same principle.
>
> **Table 1.** Ablation study on LoRA ranks $r$ across two benchmarks of varying difficulty.
> ㅤ | CIFAR-100 | ㅤ | ImageNet-R | ㅤ
> -------|-----------|-----------|-----------|-----------
> ㅤ | $\overline{A}_{10}$ | Avg. | $\overline{A}_{10}$ | Avg.
> rank=16 | 88.31 | 92.59 | 72.10 | 79.18
> rank=10 | 87.91 | 92.27 | 72.86 | 78.95
> rank=4 | 87.79 | 92.12 | 72.42 | 78.78
> rank=1 | 86.77 | 91.61 | 70.37 | 77.27
>
> To exclude the effect of additional regularization, we examine the stability–plasticity trade-off using vanilla LoRA. We observe that a vanilla LoRA with a lower rank (e.g., $r=1$) even outperforms higher-rank settings in terms of final average accuracy. The corresponding results are presented in **Table 2** and **Table 3**.
>
> The reason is that a smaller rank naturally provides the model with stronger stability, but at the cost of reduced plasticity. For CIFAR-100, this cost of plasticity is relatively minor, whereas for ImageNet-R, it becomes much more pronounced. This indicates that for more challenging benchmarks, the model should adopt a larger rank to ensure sufficient plasticity.
>
> **Table 2.** Ablation study on LoRA Ranks ($r$) in the Vanilla LoRA setting.
> CIFAR-100 | $\overline{A}_{10}$ | Avg. | Stability | Plasticity
> -------|-----------|-----------|-----------|-----------
> rank=16 | 83.05 | 89.71 | 86.25 | 100.42
> rank=10 | 82.99 | 89.74 | 87.56 | 98.86
> rank=4 | 83.35 | 90.10 | 88.70 | 98.24
> rank=1 | 83.72 | 89.94 | 89.95 | 97.36
>
> **Table 3.** Ablation study on LoRA Ranks ($r$) in the Vanilla LoRA setting.
> ImageNet-R | $\overline{A}_{10}$ | Avg. | Stability | Plasticity
> -------|-----------|-----------|-----------|-----------
> rank=16 | 66.05 | 76.26 | 74.17 | 101.41
> rank=10 | 66.32 | 76.35 | 75.69 | 99.86
> rank=4 | 67.52 | 76.33 | 77.27 | 97.30
> rank=1 | 68.43 | 77.09 | 81.95 | 96.75
>
> *Note that the plasticity is computed based on the upper bound at rank 10. Therefore, when the rank is 16, the plasticity can exceed 100%.

---

> ### Author Response · Authors · 2025-11-21
> **Response to Reviewer ETzF (Question 1.2)**
>
> ### (Q1.2) Discussion on the method's sensitivity to dataset complexity.
>
> We believe that the observed sensitivity of the method on ImageNet-R primarily stems from the dataset’s higher domain shift. This indicates that the limitation is largely related to the model’s ability to generalize under distributional changes.
>
> As shown in Tables 2 and 3 (Q1.1), a vanilla model without any mechanism to prevent forgetting can even achieve higher final average accuracy at a lower rank. This suggests that low-rank methods naturally favor stability in continual learning. For task-isolated low-rank CL methods, this inherent stability can be leveraged, and plasticity can be further enhanced by introducing task-specific low-rank modules. This may explain why EWC-LoRA performs slightly worse than these methods.
>
> For the more challenging benchmark ImageNet-R, we use Exact Fisher to estimate the changeable parameter space more accurately. The results of using Exact Fisher across four benchmarks are shown in **Table 4**. Compared to the simpler CIFAR-100, this provides greater advantages in learning new tasks, leading to more noticeable improvements in overall performance on ImageNet-R and ImageNet-A. This represents a practical strategy for addressing the limitations of regularization-based low-rank CL methods on complex target benchmarks.
>
> **Table 4.** Comparison of EWC-LoRA using Empirical Fisher and Exact Fisher.
> Tasksㅤ | CIFAR-100 | ㅤ | DomainNet | ㅤ | ImageNet-R | ㅤ | ImageNet-A | ㅤ
> -------|-----------|-----------|-----------|-----------|-----------|-----------|-----------|-----------
> ㅤ | $\overline{A}_{10}$ | Avg. | $\overline{A}_{5}$ | Avg. | $\overline{A}_{10}$ | Avg. | $\overline{A}_{10}$ | Avg.
> EWC-LoRA | 87.91 | 92.27 | 73.46 | 79.58 | 72.86 | 78.95 | 59.89 | 68.33
> EWC-LoRA-Exact | 88.28 | 92.37 | 73.48 | 79.90 | 74.54 | 80.20 | 60.73 | 69.59
>
> Nevertheless, we fully agree that automatically adapting the rank based on task complexity is a promising direction. We have added a discussion of this point in **Appendix A.3.1** (Line 1157-1180) to clarify our observations and outline this extension.

---

> ### Author Response · Authors · 2025-11-21
> **Response to Reviewer ETzF (Question 2)**
>
> ## (Question 2) Clarification on the unique advantages of EWC-LoRA.
>
> We thank the reviewer for raising this point and for the opportunity to clarify the unique advantages of EWC-LoRA.
>
> **From a storage efficiency perspective**, InfLoRA incurs a cost comparable to that of EWC-LoRA. Aside from the model itself, InfLoRA and EWC-LoRA respectively maintain a gradient subspace and a Fisher Information Matrix for previous tasks. For SD-LoRA, its variants (RR and KD) significantly enhance parameter efficiency. Both methods adopt a task-isolated LoRA strategy: RR reduces the LoRA rank for later tasks, whereas KD integrates new directions into the existing ones. KD may reach saturation when the old directions are sufficient to represent the new ones, highlighting one of the advantages of SD-LoRA. In contrast, the unique advantage of EWC-LoRA lies in its ability to control the learning direction of LoRA by identifying parameters that are less important for previous tasks, rather than explicitly freezing the old LoRA parameters. In terms of storage efficiency, the relationship among these methods is as follows: EWC-LoRA $\approx$ InfLoRA > KD > RR > SD-LoRA.
>
> **From a performance perspective**, the three methods exhibit distinct characteristics. For InfLoRA, its constraints on the subspace tend to limit the model's plasticity, as shown in the analysis of the stability–plasticity trade-off (Table 3 in the main text). In contrast, EWC-LoRA allows more flexible updates of the parameters. SD-LoRA, on the other hand, demonstrates superior plasticity across benchmarks but often places less emphasis on older tasks, particularly the first task. Our stability–plasticity analysis further reveals a unique advantage of EWC-LoRA: it achieves a better balance between stability and plasticity compared to the other two methods.

---

### Official Review · Reviewer_UxrP · 2025-11-06

**Soundness:** 3
**Presentation:** 3
**Contribution:** 2
**Rating:** 4
**Confidence:** 4

**Summary:**

This paper introduces EWC-LoRA, a parameter-efficient continual learning method that integrates Elastic Weight Consolidation (EWC) with low-rank adaptation for large-scale pre-trained models. Unlike prior low-rank approaches that rely on task-specific modules, EWC-LoRA regularizes a shared low-rank update using the Fisher Information Matrix, maintaining constant memory while improving the stability–plasticity trade-off. Experiments show that EWC-LoRA outperforms vanilla LoRA by 8.92% on average and matches or surpasses previous low-rank continual learning methods.

**Strengths:**

1. The paper is well-organized and easy to follow.

2. The investigated problem of continual learning using low-rank adaptation is important.

**Weaknesses:**

1. It would be helpful if the authors could further elaborate on the paper’s main contributions. In particular, clarifying the fundamental challenge in integrating EWC with LoRA would strengthen the work.

2. The experiments are primarily conducted on artificial image datasets. Including additional experiments on large language models (LLMs) would make the evaluation more comprehensive and convincing.

3. The paper seems to overlook an important line of research on continual learning (continual fine-tuning) of LLMs, such as O-LoRA, LoRA-MoE, and TreeLoRA. Discussing how this work relates to or differs from these studies would provide better context and positioning.

4. The performance improvements over previous methods appear relatively modest; as shown in Table 1, the accuracy increases only slightly. A more detailed analysis of these results could help clarify the practical significance of the gains.

[O-LoRA] Orthogonal Subspace Learning for Language Model Continual Learning

LoRA-MoE: Alleviating World Knowledge Forgetting in Large Language Models via MoE-Style Plugin

TreeLoRA: Efficient Continual Learning via Layer-Wise LoRAs Guided by a Hierarchical Gradient-Similarity Tree

**Questions:**

See Weaknesses above.

---

> ### Author Response · Authors · 2025-11-21
> **Response to Reviewer UxrP (Weakness 1, 2, 3)**
>
> We thank the Reviewer for the time spent on our work and the suggestions for improving its quality. Below, we address the points raised by the Reviewer, and we have made the corresponding revisions to the main text, which has greatly improved the quality of our work.
>
> ## (Weakness 1) The fundamental challenge in integrating EWC with LoRA.
>
> We appreciate the reviewer’s insightful comment and the opportunity to clarify the main contributions and the key challenge in integrating EWC with LoRA.
>
> The main contribution of this paper is that we theoretically and empirically demonstrate that a naïve integration of EWC with low-rank adaptation is suboptimal. The fundamental challenge in combining these two approaches lies in the mismatch between their parameter update mechanisms. EWC constrains parameter drift in the full-dimensional parameter space using the Fisher Information Matrix (FIM), whereas LoRA restricts updates to a low-rank subspace. Computing FIMs separately for the low-rank matrices and updating them individually imposes overly strong constraints on the model’s updates, since each element of the overall update $\Delta \mathbf{W}$ depends on the joint product—thereby reducing the model’s plasticity, as empirically demonstrated in our ablation study (Table 6).
>
> We have revised **Section 1** (Introduction Line 071-072, 091-093) and **Section 3** (Methodology Line 212-213) to better highlight the contribution and to clarify the above challenge. We believe these revisions make our contributions clearer and strengthen the motivation of our work.
>
> ## (Weakness 2) Experiments on large language models (LLMs).
>
> We thank the reviewer for this valuable suggestion. We agree that evaluating our method on LLMs can further demonstrate its effectiveness.
>
> Following the reviewer’s suggestion, we extended our experiments to language continual learning (CL) benchmarks and compared EWC-LoRA with two LoRA-based baselines (O-LoRA and TreeLoRA) on the standard language CL tasks used in O-LoRA. We report the per-task accuracy across three task orders under two pretrained models in **Table 1** (LLaMa-3.2-1B-Instruct) and **Table 2** (T5-large). The results demonstrate that EWC-LoRA is also effective and applicable to LLMs on language CL benchmarks.
>
> We have added the results to **Table 3** along with the corresponding discussion in **Section 4.2** (Line 376-379). We also visualize the performance throughout the training process in **Figure 6** and **7** in the **Appendix A.3.1**. For the T5-large pretrained model, we only compare against O-LoRA, as TreeLoRA requires substantially more GPU memory, making it difficult for us to reproduce its results.
>
> **Table 1.** Comparison results on the standard CL benchmark with the LLaMa-3.2-1B-Instruct model.
> Task Order | Method | Task 1 | Task 2 | Task 3 | Task 4 | Avg.
> -------|-----------|-----------|-----------|-----------|-----------|-----------
> Order 1 | O-LoRA | 44.85 | 47.47 | 55.64 | 79.88 | 56.96
> ㅤ | TreeLoRA | 47.20 | **49.25** | 55.92 | **81.78** | 58.54
> ㅤ | EWC-LoRA | **69.92** | 47.55 | **56.94** | 70.55 | **61.17**
> Order 2 | O-LoRA | **69.57** | 36.24 | 56.41 | 60.74 | 55.74
> ㅤ | TreeLoRA | 69.55 | 44.33 | 54.24 | 59.72 | 56.96
> ㅤ | EWC-LoRA | 69.52 | **46.25** | **56.72** | **69.38** | **60.47**
> Order 3 | O-LoRA | **55.77** | 51.38 | 72.85 | 89.26 | 67.32
> ㅤ | TreeLoRA | 55.40 | 48.33 | 69.41 | 88.52 | 65.42
> ㅤ | EWC-LoRA | 53.68 | **51.64** | **75.73** | **89.39** | **67.61**
>
> **Table 2.** Comparison results on the standard CL benchmark with the T5-large model.
> Task Order | Method | Task 1 | Task 2 | Task 3 | Task 4 | Avg.
> -------|-----------|-----------|-----------|-----------|-----------|-----------
> Order 1 | O-LoRA | 93.93 | 55.46 | 65.14 | **88.22** | 75.69
> ㅤ | EWC-LoRA | **97.05** | **56.29** | **71.03** | 87.68 | **78.01**
> Order 2 | O-LoRA | 95.00 | 50.00 | 83.93 | 70.76 | 74.92
> ㅤ | EWC-LoRA| **96.78** | **51.22** | **86.79** | **72.61** | **76.85**
> Order 3 | O-LoRA | 64.39 | **52.46** | 82.12 | 98.64 | **74.40**
> ㅤ | EWC-LoRA | **65.96** | 48.63 | **83.89** | **98.70** | 74.30
>
> ## (Weakness 3) Discussion on continual learning on LLMs.
>
> We thank the reviewer for pointing this out, as it helps better distinguish our work from prior works.
>
> Both O-LoRA and EWC-LoRA belong to regularization-based methods. However, O-LoRA mitigates task interference by enforcing geometric orthogonalization in the update subspace, which protects the direction of updates, whereas EWC-LoRA protects the magnitude of updates through Fisher-based penalties. LoRA-MoE assigns different LoRA experts to different tasks to prevent forgetting, while TreeLoRA avoids forgetting by selecting task-shared and task-specific structures. Both methods, therefore, belong to the task-structure isolation methods.
>
> We have added these three works to **Section 2.** Related Works, and we have especially clarified the distinctions between O-LoRA and our method.

---

> ### Author Response · Authors · 2025-11-21
> **Response to Reviewer UxrP (Weakness 4)**
>
> ## (Weakness 4) Practical significance of the performance gains presented in Table 1.
>
> We thank the reviewer for pointing out the need to articulate the practical significance of the results in the main Table. We would first like to clarify the following adjustment to avoid confusion: Due to updates in the paper’s tables, the Table 1 referenced in your original question is labeled as Table 2 in the current version.
>
> (1) First, EWC-LoRA demonstrates the effectiveness of applying regularization in low-rank adaptation. Compared with vanilla LoRA, the gain is substantial and varies from 3.67-19.88% on the four datasets. This also suggests that other low-rank CL methods may similarly benefit from incorporating EWC in practice.
>
> (2) When compared with the state-of-the-art, it is more modest, but on DomainNet EWC-LoRA reduces the gap with the upper bound Joint Training significantly from 5.57% of SD-LoRA to 3.38% for EWC-LoRA (and further to 3.36% with exact Fisher computation). On the challenging ImageNet-A, it reduces the gap considerably from 7.39% for CL-LoRA to only 5.12% for EWC-LoRA (and further to 4.28% with exact Fisher computation). We have added experimental results using Exact Fisher on four datasets, as shown in Table 3.
>
> (3) We would like to highlight an important conclusion drawn from both Table 2 and Table 4. Even when two CL methods achieve similar average accuracy, they can differ substantially in their stability–plasticity trade-off, a crucial aspect that has often been overlooked in prior work.
>
> We have also added this discussion to the corresponding **Section 4.2** (Line 370-373).
>
> **Table 3.** Comparison results on four benchmarks.
> Tasksㅤ | CIFAR-100 | ㅤ | DomainNet | ㅤ | ImageNet-R | ㅤ | ImageNet-A | ㅤ
> -------|-----------|-----------|-----------|-----------|-----------|-----------|-----------|-----------
> ㅤ | $\overline{A}_{10}$ | Avg. | $\overline{A}_{5}$ | Avg. | $\overline{A}_{10}$ | Avg. | $\overline{A}_{10}$ | Avg.
> Vanilla LoRA | 82.99 | 89.74 | 69.79 | 77.44 | 64.87 | 75.57 | 40.01 | 58.28
> EWC-LoRA | 87.91 | 92.27 | 73.46 | 79.58 | 72.86 | 78.95 | 59.89 | 68.33
> EWC-LoRA-Exact | 88.28 | 92.37 | 73.48 | 79.90 | 74.54 | 80.20 | 60.73 | 69.59

---

> > ### Comment · Reviewer_UxrP · 2025-11-25
> >
> > Thanks to the author for the response, which addressed my concerns. I have therefore decided to raise my score.

---

> > > ### Author Response · Authors · 2025-11-26
> > >
> > > Dear reviewer UxrP,
> > >
> > > Your thoughtful consideration of our responses is sincerely appreciated. We welcome any further questions or discussions and appreciate the opportunity to enhance our work.
> > >
> > > Thank you once again for your valuable time and effort!
> > >
> > > Best regards,
> > > The Authors

---

### Author Response · Authors · 2025-12-02
**Message to the Reviewers and Area Chair**

## **Message to the Reviewers and Area Chair:**

We sincerely appreciate your time and consideration of our submission. **Following our rebuttal, we are grateful that all four reviewers supported acceptance.** In this global response, we respectfully summarize the key concerns raised by the reviewers and highlight the clarifications and evidence we have provided in the individual rebuttals.

- **Main contribution and practical significance:** We demonstrate that a naïve combination of EWC and low-rank adaptation (LoRA) is inherently suboptimal, and that our proposed EWC-LoRA provides clear benefits in balancing stability and plasticity. *The corresponding updates have been made primarily in Sections 1, 3, and 4.2 in response to reviewers #UxrP, #ETzF, and #gUkm*.

- **Effectiveness of EWC-LoRA on LLMs and comparison with LoRA-based vision and language CL tasks:** Newly added experiments show that EWC-LoRA is also effective on standard language CL tasks, and consistently shows reliable performance across both vision and language benchmarks. *These new results have been incorporated into Sections 2 and 4.2 in response to reviewers #UxrP and #gUkm*.

- **Discussion of limitations of regularization-based low-rank CL:** We analyzed the accuracy of parameter-importance estimation as learning progresses, showing that observed inaccuracies are primarily due to changes in magnitude and directional scaling, while the key parameters identified by EWC remain largely stable. *This discussion has been added to Appendix A.3.2 in response to reviewers #ETzF and #VVBD*.

- **Additional experimental analyses:** Beyond addressing the key concerns above, we have included further analyses in response to reviewers #gUkm, #ETzF, and #VVBD. These analyses examine the effect of the parameter $\gamma$, the impact of the LoRA rank $r$, and training-time comparisons. *The corresponding supplementary results have been added to the Appendix*.

---

We once again thank all reviewers and the Area Chair for their careful evaluation of our submission. We sincerely hope that our responses have effectively addressed the concerns raised. **We are grateful to reviewer #UxrP for kindly raising the score from 4 to 6, which is greatly encouraging to us. We deeply appreciate the positive feedback provided by all reviewers.**

---

### Meta-Review · Area_Chair_UTZN · 2026-01-07

**Summary:**

This paper revisits weight regularization in the context of parameter-efficient continual learning, learning a shared low-rank update rather than allocating task-specific LoRA modules, keeping the storage requirement constant for the number of tasks. The paper further provides a systematic investigation showing that naïvely applying EWC under low-rank parameterization can be suboptimal, and proposes EWC-LoRA, which estimates parameter importance over the full-dimensional update space while still training via low-rank updates. Extensive experiments across multiple benchmarks show consistent gains, including an average improvement over vanilla LoRA, while achieving competitive performance relative to state-of-the-art low-rank CL baselines, with an explicit analysis of the stability–plasticity trade-off.
Overall, the reviews are generally positive. Reviewers find the work technically sound, clearly written, and well motivated for continual adaptation of large pre-trained models. Two reviewers particularly value the paper’s principled analysis of why “naïve EWC + LoRA” is mismatched, and the empirical evaluation emphasizing stability/plasticity rather than only final average accuracy.

**Reviewer Concerns:**

UxrP: Initially raised concerns about novelty/positioning and missing related work in LLM continual adaptation; also questioned whether gains are large enough. They acknowledged the overall clarity and the importance of revisiting regularization in PECL, but wanted stronger contextualization and evidence.

ETzF: Positive; viewed the approach as effective and the evaluation convincing, while noting Fisher-related overhead and that originality is moderate but still meaningful in this emerging PECL-CL setting.

VVBD: Positive; highlighted solid methodology and experimental validation, but asked for clarification on Fisher estimation quality over long sequences and practical behavior under harder/longer task streams.

gUkm: Moderately positive; agreed the method is sound and the ablations are fairly complete, but expressed that the conceptual novelty is closer to a careful adaptation/validation of EWC principles for LoRA-style continual tuning.

**Reviewer Scores:**

A recurring concern is that the method may appear as an adaptation of EWC to LoRA. I agree the high-level ingredients are familiar; however, the paper’s contribution is not merely “apply EWC,” but to (i) identify and formalize why naïve regularization in low-rank parameterizations can diverge from full-space regularization, and (ii) propose a principled fix that regularizes in the full-dimensional update subspace while retaining PEFT efficiency, plus a shared-module formulation with constant memory across tasks.
Reviewers questioned whether Fisher estimation introduces practical overhead and whether empirical Fisher degrades with longer task sequences. The paper addresses this with explicit discussion/experiments comparing Fisher estimation strategies, including computing Exact Fisher on a small batch to reduce overhead, and it transparently states the degradation issue as a limitation and points to rehearsal-based Fisher estimation as a mitigation.
One reviewer asked for stronger positioning against concurrent/adjacent LoRA-based continual adaptation lines. While this is a fair request, the paper’s experimental scope across multiple benchmarks, along with its focus on shared-module regularization vs. task-specific modules, makes the contribution reasonably distinct; the rebuttal/updates (as reflected in the added analyses/ablations) further clarify when and why the method helps.

---

### Decision · Program_Chairs · 2026-01-26

Accept (Poster)